# Thioredoxin-1 distinctly promotes NF-κB target DNA binding and NLRP3 inflammasome activation independently of Txnip

Jonathan Muri, Helen Thut, Qian Feng, Manfred Kopf*

Institute of Molecular Health Sciences, ETH Zürich, Zürich, Switzerland

**Abstract** Antioxidant systems, such as the thioredoxin-1 (Trx1) pathway, ensure cellular redox homeostasis. However, how such systems regulate development and function of myeloid cells is barely understood. Here we show that in contrast to its critical role in T cells, the murine Trx1 system is dispensable for steady-state myeloid-cell hematopoiesis due to their capacity to tap the glutathione/glutaredoxin pathway for DNA biosynthesis. However, the Trx1 pathway instrumentally enables nuclear NF-κB DNA-binding and thereby pro-inflammatory responses in monocytes and dendritic cells. Moreover, independent of this activity, Trx1 is critical for NLRP3 inflammasome activation and IL-1β production in macrophages by detoxifying excessive ROS levels. Notably, we exclude the involvement of the Trx1 inhibitor Txnip as a redox-sensitive ligand of NLRP3 as previously proposed. Together, this study suggests that targeting Trx1 may be exploited to treat inflammatory diseases.

*For correspondence:
manfred.kopf@biol.ethz.ch

## Introduction

Aerobic organisms must continuously prevent the oxidation of their cellular macromolecules. To achieve this, they have evolved a sophisticated system composed of antioxidant proteins that ensure redox homeostasis in the cell. The two main pathways that make up this cellular redox system are the thioredoxin (Trx) system and the glutathione (GSH) system (*Arnér and Holmgren, 2000*; *Couto et al., 2016*). GSH can donate electrons to both glutaredoxins (Grx) and glutathione peroxidases (Gpx), a member of which, namely Gpx4, has been recently shown to prevent ferroptotic cell death in T and B1 cells (*Matsushita et al., 2015*; *Muri et al., 2019a*). Both the Trx and GSH systems ultimately rely on the reducing power from NADPH, which in mammals is generated by the alternative oxidation of glucose in the pentose phosphate pathway (PPP) (*Stanton, 2012*). In turn, the Trx and the GSH/Grx systems provide reducing equivalents that sustain a number of cellular processes crucial for cell function, cell survival, cell proliferation, and redox-regulated signaling pathways. For instance, they fuel the reducing power for DNA synthesis by ribonucleotide reductase (RNR) during proliferation, for scavenging reactive oxygen species (ROS), and for the reduction of protein disulfides back to their reduced state (*Camier et al., 2007*; *Holmgren, 1985*; *Holmgren, 2000*; *Muri et al., 2018*; *Potamitou et al., 2002*; *Prinz et al., 1997*). In the cytosolic Trx1 system, Trx reductase 1 (TrxR1, encoded by the *Txnrd1* gene) has the unique capacity to transfer electrons from NADPH to oxidized Trx1 (encoded by the *Txn1* gene), thereby keeping Trx1 in its reduced state. Thioredoxin-interacting protein (Txnip) is an additional member of the Trx1 system, which negatively regulates Trx function (*Arnér, 2009*; *Mustacich and Powis, 2000*). In the GSH/Grx system, by contrast, glutathione reductase (Gsr) maintains the pool of cellular GSH in its reduced state, which in turn further reduces oxidized Grx (*Lu, 2013*). To which extent the Trx and the GSH/glutaredoxin systems compensate for each others functions in vivo remains unknown.

Macrophages and dendritic cells (DCs) secrete several inflammatory cytokines to orchestrate immune responses. Upon sensing microbial components via Toll-like receptors (TLR), they utilize the MyD88 adaptor to activate nuclear factor-κB (NF-κB)-dependent transcription of pro-inflammatory cytokines including IL-6 (encoded by the *Il6* gene), IL-12p40 (encoded by the *Il12b* gene), TNF-α (encoded by the *Tnfa* gene) and IL-1β (encoded by the *Il1b* gene) (*Akira and Takeda, 2004*). Secretion of IL-1β, however, needs a second signal required for inflammasome assembly, caspase-1 or −11 activation, processing of the immature IL-1β precursor (pro-IL-1β), and subsequent release of the active and mature form of IL-1β (*Martinon et al., 2002*). A variety of different stimuli that activate inflammasome have been described in the field, especially for the canonical NLRP3 inflammasome (*Broz and Dixit, 2016*). Interestingly, cellular redox regulation and ROS production have been described to regulate both NF-κB activity (*Morgan and Liu, 2011*) and NLRP3 inflammasome function (*Tschopp and Schroder, 2010*). However, the molecular mechanisms of this redox regulation remain to be defined. In particular, the Trx-inhibitor Txnip has been proposed to activate the NLRP3 inflammasome in response to ROS (*Zhou et al., 2010*), although these results remain controversial (*Masters et al., 2010*). Therefore, the mechanism by which redox regulation is linked to NF-κB and inflammasome regulation is not fully resolved yet.

We have previously characterized the roles of the Trx1 and GSH/Grx1 systems in T- and B-cell immunity. Notably, we demonstrated that the Trx1 system is critically required to fuel reducing power for the sustainment of DNA biosynthesis during metabolic reprogramming in T but not in follicular B cells (*Muri et al., 2018*; *Muri et al., 2019b*). In the present study, we found that the Trx1 system is dispensable for the steady-state hematopoiesis of myeloid cells (i.e. neutrophils, monocytes, macrophages and DC subsets), which efficiently rearrange their redox system toward the GSH/Grx pathway to fuel proliferation when the Trx1 system is absent. Furthermore, we demonstrated how the Trx1 and Grx systems differentially regulate the inflammatory responses of bone marrow-derived DCs (BMDCs) and macrophages (BMDMs). Specifically, while the first utilize the reducing power of the Trx1 system to allow efficient NF-κB p65 transcription factor binding to its DNA response element, the latter need Trx1-dependent antioxidant functions to enable NLRP3 inflammasome formation and IL-1β release. Importantly, our data exclude a role of Txnip in NLRP3 inflammasome regulation as previously proposed (*Zhou et al., 2010*). In conclusion, these results suggest that therapeutic intervention aimed at blocking the Trx1 system may be beneficial to treat inflammatory diseases.

## Results

### The Trx1 system is dispensable for myeloid-cell but not T-cell development and homeostatic maintenance

To investigate the requirement of the Trx1 system in myeloid cells during development and homeostatic maintenance, we crossed mice carrying tamoxifen (TAM)-inducible Rosa26-CreERT2 with mice carrying *loxP*-flanked *Txnrd1* alleles to generate progeny (*Txnrd1*^fl/fl^;Rosa26-CreERT2) in which *Txnrd1* is globally deleted upon TAM administration. Cre-mediated deletion in total bone marrow cells and in CD11b^+^ splenocytes of *Txnrd1*^fl/fl^;Rosa26-CreERT2 mice was complete at the genomic DNA and mRNA levels (*Figure 1—figure supplement 1*). TAM injection into *Txnrd1*^fl/fl^;Rosa26-CreERT2 mice completely abolished the development of T cells in the thymus, leading to a massive reduction of CD4^+^CD8^+^ double positive, CD4^+^ and CD8^+^ single positive thymocytes (*Figure 1A*), as shown previously (*Muri et al., 2018*). In striking contrast, numbers of eosinophils, neutrophils, monocytes and DCs were comparable in the bone marrow of TAM-treated *Txnrd1*^fl/fl^;Rosa26-CreERT2 and control (*Txnrd1*^fl/fl^) mice (*Figure 1B* and *Figure 1—figure supplement 2A*), indicating that the Trx1 system is dispensable for the process of hematopoiesis in these cells. Despite their short lifespan, monocytes, neutrophils and eosinophils were also present in the blood with a similar frequency in the absence and presence of *Txnrd1* (*Figure 1C* and *Figure 1—figure supplement 2B*). Moreover, *Txnrd1* deficiency also did not affect total numbers of alveolar macrophages, eosinophils, neutrophils, monocytes and conventional type 1 and 2 DCs (cDC1 and cDC2) in the lungs (*Figure 1D* and *Figure 1—figure supplement 2C*). Similarly, these populations were also unchanged in the spleen apart from a reduction in total numbers of cDC2 (*Figure 1E* and *Figure 1—figure supplement 2D*). Taken together, these results demonstrate that, in contrast to its critical role in T cells,

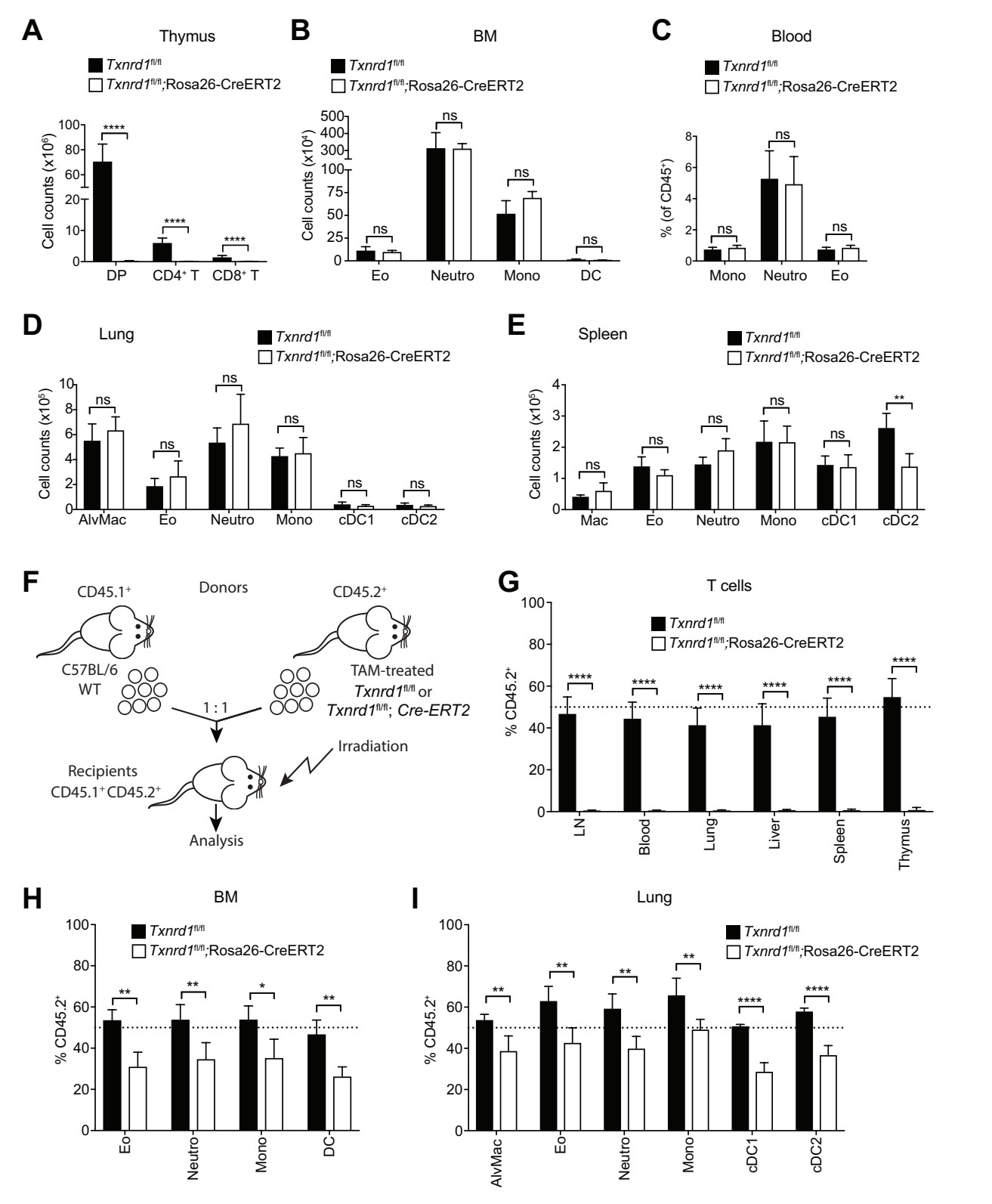

**Figure 1.** The Trx1 system is largely dispensable for the development and homeostatic maintenance of myeloid cells. (A–E) *Txnrd1*[fl/fl];Rosa26-CreERT2 mice and *Txnrd1*[fl/fl] littermates were injected with TAM to delete the *Txnrd1* gene and were analyzed by flow cytometry 2 weeks later. Depicted are the total numbers or percentages of the indicated populations in the thymus (A), bone marrow (BM; B), blood (C), lungs (D), spleen (E); n = 4–5 mice). (F–I) Lethally irradiated WT mice (CD45.1+CD45.2+) were reconstituted with a 1:1 mixture of WT (CD45.1+) and TAM-treated *Txnrd1*[fl/fl];Rosa26-CreERT2

*Figure 1 continued on next page*

*Figure 1 continued*

(CD45.2$^+$) bone marrows (or *Txnrd1*$^{fl/fl}$ as control). After reconstitution, the percentage of donor CD45.2$^+$ cells among the indicated cell populations was determined by flow cytometry (n = 4–5 mice). (F) Schematic showing the experimental setup. (G) Depicted are the CD45.2$^+$ percentages of total TCRβ$^+$ T cells in the indicated organs. (H, I) The percentages of CD45.2$^+$ cells among the indicated myeloid populations in the bone marrow (BM; H) and lungs (I) are shown. DP, CD4$^+$CD8$^+$ double positive thymocytes; CD4$^+$ T, CD4$^+$ single positive thymocytes; CD8$^+$ T, CD8$^+$ single positive thymocytes; Eo, eosinophils; Neutro, neutrophils; Mono, monocytes; DC, dendritic cells; AlvMac, alveolar macrophages; cDC1/2, type 1/2 conventional dendritic cells; Mac, macrophages; LN, lymph nodes. Bar graphs show mean + standard deviation (A–E, G–I). Data are representative of three independent experiments. For each panel, a representative experiment with biological replicates (A–E,G–I) is shown. Student's *t* test (two-tailed, unpaired) was used for the comparison of two groups (A–E, G–I): *, p≤0.05; **, p≤0.01; ***, p≤0.001; ****, p≤0.0001; ns, not significant. The online version of this article includes the following figure supplement(s) for figure 1:

**Figure supplement 1.** Efficiency of *Txnrd1* gene deletion.
**Figure supplement 2.** Gating strategies for the analysis of distinct myeloid-cell populations in vivo.
**Figure supplement 3.** The Trx1 system is largely dispensable for the maintenance and homeostasis of myeloid cells in mixed-bone marrow chimera settings.

the Trx1 system is dispensable for the development and the homeostatic maintenance of various types of myeloid-cell populations.

## *Txnrd1*-deficient bone marrow can partially refill the myeloid but not the T-cell compartment of irradiated recipients

Since distinct myeloid populations were not reduced in the absence of the Trx1 system at the steady-state, we next investigated whether bone marrow lacking *Txnrd1* could efficiently reconstitute the hematopoietic compartment of irradiated wild type (WT) hosts in a competitive situation with WT cells. To test this, we generated mixed-bone marrow chimeras by reconstituting the hematopoietic compartment of irradiated C57BL/6 mice (CD45.1$^+$CD45.2$^+$) with an equal ratio of congenically marked donor bone marrow cells from WT (CD45.1$^+$) and TAM-treated *Txnrd1*$^{fl/fl}$; Rosa26-CreERT2 (CD45.2$^+$) or *Txnrd1*$^{fl/fl}$ (CD45.2$^+$) mice as control (*Figure 1F*). Expectedly, *Txnrd1*-deficient bone marrow completely failed to refill the T-cell compartment of irradiated recipients (*Figure 1G*), consistent with our previous findings reporting a crucial role of Trx1 for T-cell proliferation (*Muri et al., 2018*). Notably, eosinophils, neutrophils, monocytes, DCs and macrophages lacking *Txnrd1* were reduced by 30–50% in the bone marrow, blood, spleen and lungs, as compared to WT counterparts, thus indicating a small contribution of *Txnrd1* for the expansion of myeloid-cell precursors that could be revealed in a competitive situation with WT cells (*Figure 1H,I* and *Figure 1—figure supplement 3A,B*), similar to B-cell development (*Figure 1—figure supplement 3C*; *Muri et al., 2019b*). Together, these results confirm that, in contrast to its crucial role during thymic T-cell development, the Trx1 pathway is largely dispensable for the development and the maintenance of myeloid cells.

## The GSH/Grx system sustains steady-state hematopoiesis of myeloid cells lacking the Trx1 system

To investigate whether the GSH/Grx system may compensate for the absence of the Trx1 system, thereby allowing normal development and homeostatic maintenance of myeloid cells, we took advantage of L-buthionine-sulfoximine (BSO), which is known to deplete GSH levels upon oral administration in vivo (*Watanabe et al., 2003*). Therefore, we first generated mixed-bone marrow chimeras as described above. After reconstitution, mice were injected with TAM to delete the *Txnrd1* gene and subsequently treated with BSO over a period of 22 days to deplete GSH (*Figure 2A*). Expectedly, BSO-treated mice showed a significant reduction in total GSH levels compared to controls in the bone marrow and in the spleen (*Figure 2B,C*). In keeping with the data presented above, TAM administration led to a partial defect displayed by *Txnrd1*-deficient blood neutrophils, eosinophils and monocytes in a competitive setting with WT cells (*Figure 2D,E* and *Figure 2—figure supplement 1*). Interestingly, however, additional depletion of GSH strikingly reduced *Txnrd1*-deficient granulocytes within 5 days after oral BSO administration (*Figure 2D,E* and *Figure 2—figure supplement 1*). Neutrophils, eosinophils and monocytes lacking the Trx1 system from the bone marrow, lungs and spleen also displayed a similar sensitivity to GSH depletion as observed in the blood (*Figure 2F–H*). Although DCs were similarly affected, *Txnrd1*-deficient tissue resident

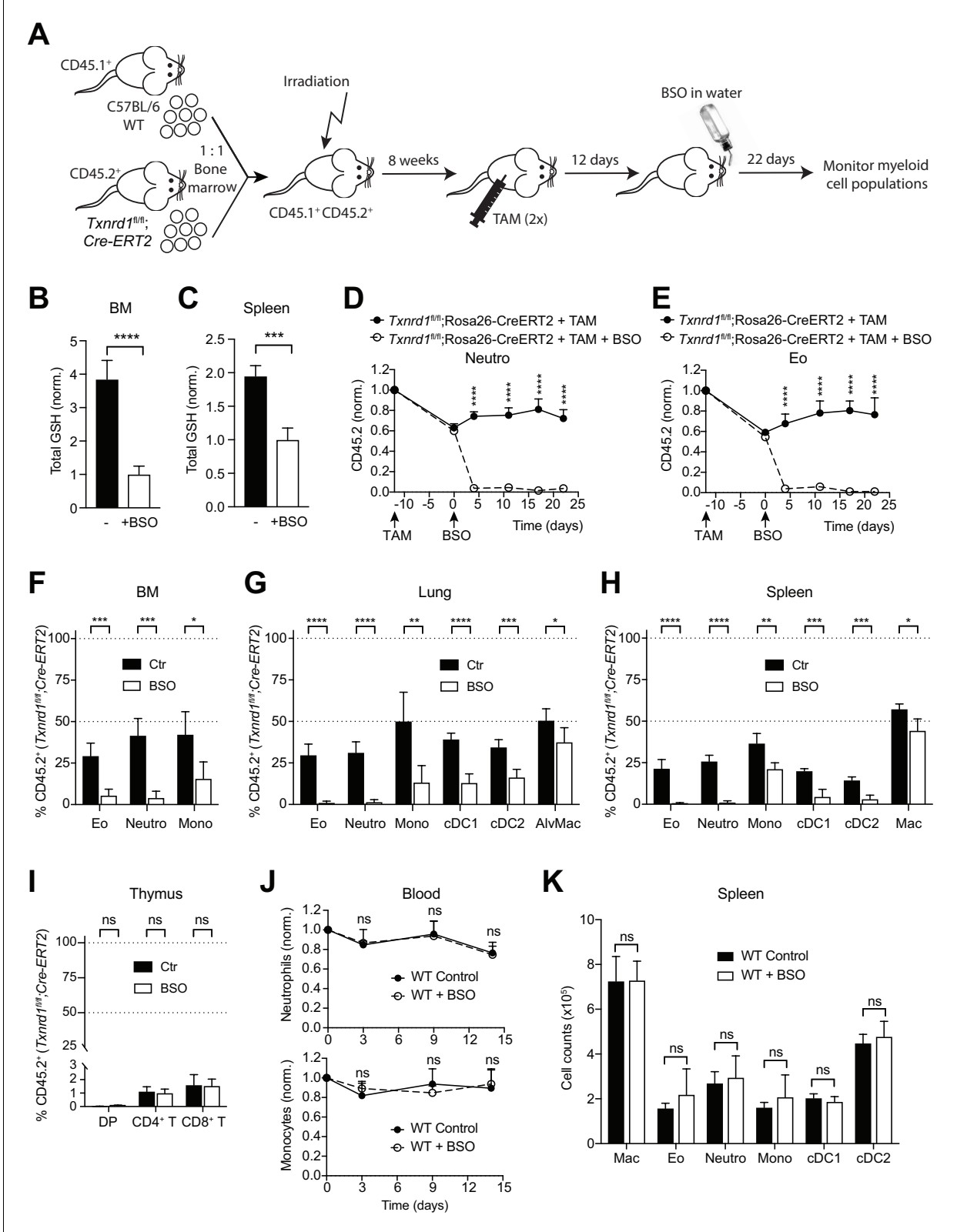

**Figure 2.** The GSH/Grx system sustains the development and maintenance of *Txnrd1*-deficient myeloid cells. (A–I) Lethally irradiated WT mice (CD45.1+CD45.2+) were reconstituted with a 1:1 mixture of WT (CD45.1+) and *Txnrd1fl/fl*;Rosa26-CreERT2 (CD45.2+) bone marrows. After reconstitution, mice were injected with TAM to delete the *Txnrd1* gene, and after 12 days BSO was administered in the drinking water to deplete GSH levels. Cell populations in the blood were monitored over time, and animals were analysed on day 22 upon BSO administration (n = 4–5 mice). (A) Schematic
*Figure 2 continued on next page*

*Figure 2 continued*

showing the experimental setup. (B, C) Depicted are the total glutathione (GSH) levels in the lysates from total bone marrow (BM) cells (B) and spleen (C). (D, E) The percentages of CD45.2$^+$ neutrophils (D) and eosinophils (E) in the blood were monitored over the indicated period of time. (F–H) The percentages of the indicated myeloid-cell populations coming from the *Txnrd1$^{fl/fl}$*;Rosa26-CreERT2 (CD45.2$^+$CD45.1$^-$) donors were analyzed in the bone marrow (BM; F), lungs (G) and spleen (H) on day 22 upon BSO administration. (I) The percentages of the indicated thymocyte populations coming from the *Txnrd1$^{fl/fl}$*;Rosa26-CreERT2 (CD45.2$^+$CD45.1$^-$) donors were analyzed 22 days after BSO administration. (J, K) WT mice were treated with BSO in the drinking water and analyzed 2 weeks later (n = 4–5 mice). (J) Total neutrophils (above) and monocytes (below) in the blood were monitored over the period of 2 weeks. The percentages at the indicated times were normalized with the percentage on day 0. (K) Shown are the total numbers of the indicated myeloid-cell populations in the spleen 2 weeks after BSO administration. Neutro, neutrophils; Eo, eosinophils; Mono, monocytes; AlvMac, alveolar macrophages; cDC1/2, type 1/2 conventional dendritic cells; Mac, macrophages; DP, CD4$^+$CD8$^+$ double positive thymocytes; CD4$^+$ T, CD4$^+$ single positive thymocytes; CD8$^+$ T, CD8$^+$ single positive thymocytes. Bar graphs and dot plots show mean + standard deviation (B–K). Data are representative of two independent experiments. For each panel, a representative experiment with biological replicates (B–K) is shown. Student's *t* test (two-tailed, unpaired) was used for the comparison of two groups (B, C, F–I, K): *, p≤0.05; **, p≤0.01; ***, p≤0.001; ****, p≤0.0001; ns, not significant. Two-way ANOVA adjusted by Bonferroni's multiple comparison test was used in D, E, J: ****, p≤0.0001; ns, not significant.

The online version of this article includes the following figure supplement(s) for figure 2:

**Figure supplement 1.** The GSH/Grx system compensates for the absence of the Trx1 pathway in *Txnrd1*-deficient monocytes.
**Figure supplement 2.** BSO administration does not affect the maintenance and homeostasis of WT myeloid cells.

macrophages in the lungs and in the spleen only showed a minor reduction upon BSO administration, consistent with their fetal origin and local persistence by slow turnover during adult life (*Figure 2G,H*). In keeping with the data presented above, *Txnrd1*-deficient CD4$^+$CD8$^+$ double positive, CD4$^+$ and CD8$^+$ single positive thymocytes were completely outcompeted by WT cells irrespective of BSO treatment (*Figure 2I*), consistent with the critical role of the Trx1 system and dispensability of the GSH/Grx system for T-cell proliferation (*Muri et al., 2018*). We further observed that oral administration of BSO to WT mice did not affect numbers of blood neutrophils and monocytes over time (*Figure 2J*). Moreover, the numbers of various myeloid populations in the bone marrow, spleen and lungs were not affected by GSH depletion in WT mice (*Figure 2K* and *Figure 2— figure supplement 2*), therefore suggesting that the GSH/Grx system is only utilized as a backup system to compensate for the absence of the Trx1 pathway. Overall, these data demonstrate that in striking contrast to T cells, myeloid cells can tap both the Trx and GSH/Grx pathways to sustain thiol-based redox reactions during development and homeostatic maintenance.

## Bone marrow expansion during emergency hematopoiesis requires the Trx1 system

Since the Trx1 system was dispensable for steady-state hematopoiesis of myeloid cells, we next investigated whether *Txnrd1*-deficient cells could also efficiently expand during emergency myelopoiesis, where proliferative stress in the bone marrow and blood neutrophilia are induced by a sublethal dose of LPS (*Boettcher et al., 2012*). To test this, we intraperitoneally administered LPS twice in a 48 hr interval to TAM-treated *Txnrd1$^{fl/fl}$* and *Txnrd1$^{fl/fl}$*;Rosa26-CreERT2 mice before sacrificing them for analysis 24 hr later (*Figure 3A*). While both groups of mice increased the numbers of CD11b$^+$Ly-6G$^-$Ly-6C$^{int}$bone marrow precursors upon LPS administration, *Txnrd1*-deficient mice displayed a limited capacity (*Figure 3B,C*). Moreover, accumulation of neutrophils in peripheral blood was also impaired in the absence of the Trx1 system (*Figure 3D,E*). Overall, these results demonstrate that the GSH/Grx pathway does not efficiently compensate for the absence of the Trx1 system during emergency hematopoiesis when massive proliferation occurs due to LPS stress. The observed defect in expansion was therefore comparable to the one shown in competitive chimera settings.

## The Trx1 system is critical for the transcription of pro-inflammatory cytokines in BMDCs

So far, we have shown that the function of the Trx1 system in various myeloid cells unlike in T cells is largely dispensable for maintenance and homeostasis, and that it can be efficiently substituted by the GSH/Grx pathway. Next, we aimed at addressing whether the Trx1 system is important for the immune function of myeloid cells, such as the secretion of pro-inflammatory cytokines. Bone marrow harvested from TAM-treated *Txnrd1$^{fl/fl}$*;Rosa26-CreERT2 and *Txnrd1$^{fl/fl}$* littermate control mice was differentiated in the presence of granulocyte-macrophage colony-stimulating factor (GM-CSF) to

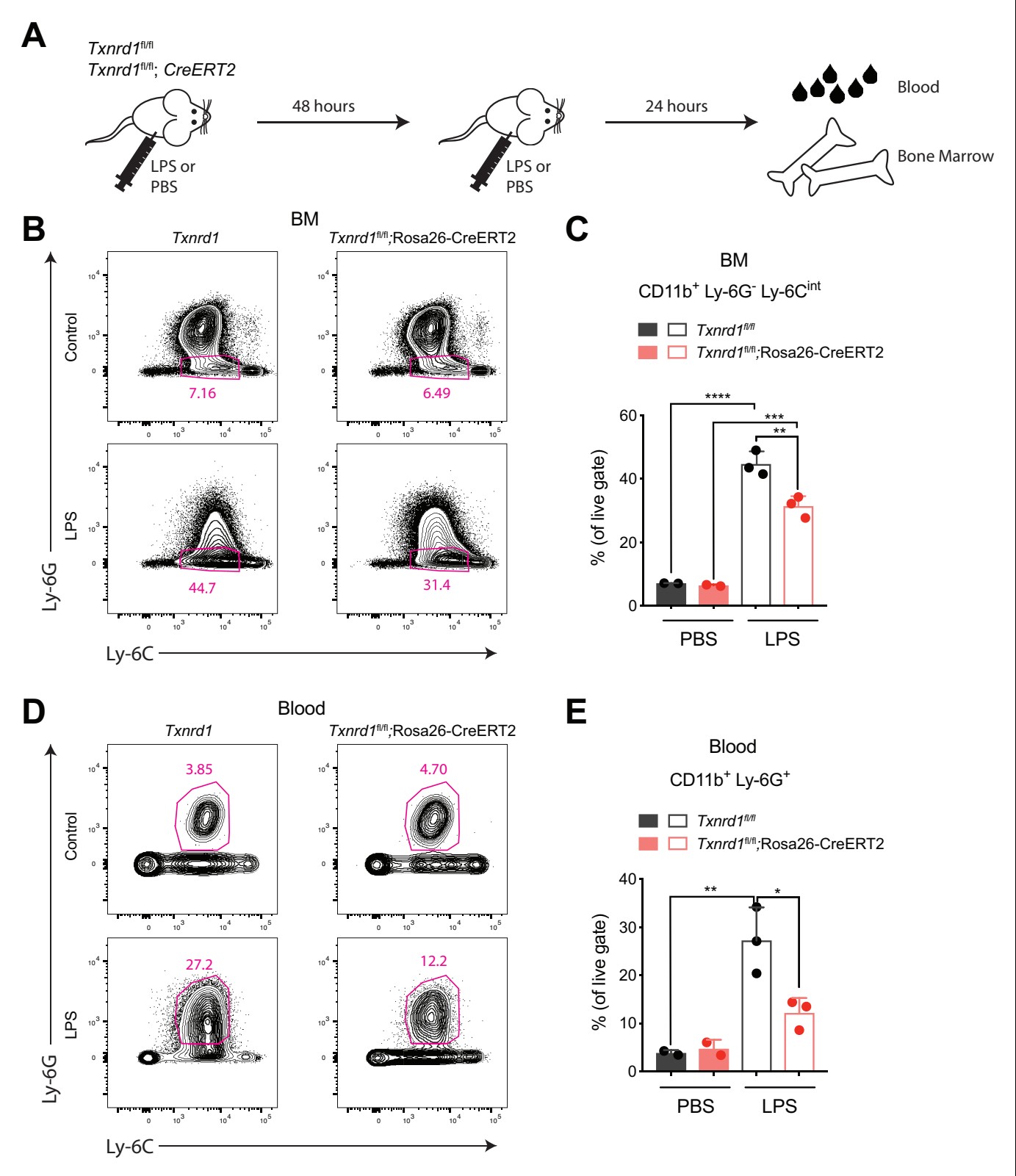

**Figure 3.** The Trx1 system is required in the bone marrow during emergency hematopoiesis. (A–E) TAM-treated *Txnrd1*[fl/fl];Rosa26-CreERT2 mice (or *Txnrd1*[fl/fl] littermate controls) were intraperitoneally injected with 35 μg LPS in a 48 hr interval and sacrificed 24 hr later for analysis (n = 3 mice). (A) Schematic showing the experimental setup. (B, C) Shown are the gating strategy (B) and percentages (C) of CD11b[+]Ly-6G[-]Ly-6C[int] cells in the bone marrow (BM). (D, E) Depicted are the gating strategy (D) and percentages (E) of CD11b[+]Ly-6G[+] neutrophils in the blood. Numbers in the FACS plots

*Figure 3 continued on next page*

*Figure 3 continued*

indicate the average percentages of the depicted gate. Bar graphs show mean + standard deviation (**C, E**). Data are representative of three independent experiments. For each panel, a representative experiment with biological replicates (**B–E**) is shown. One-way ANOVA adjusted by Tukey's multiple comparison test was used in **C, E**: *$p \leq 0.0332$; **$p \leq 0.0021$; ***$p \leq 0.0002$; ****$p \leq 0.0001$.

obtain BMDCs in culture. We first confirmed that *Txnrd1* deletion was complete at the mRNA level in knockout BMDCs (*Figure 4A*) and that cell survival was not compromised (*Figure 4B*). To study inflammatory responses, we stimulated *Txnrd1*-deficient and -sufficient BMDCs with various TLR ligands, namely CpG (TLR9 agonist), LPS (TLR4 agonist), LTA (TLR2 agonist), R837 (TLR7 agonist) and zymosan (TLR2 and dectin-1 agonist), and subsequently assessed cytokine secretion via ELISA. Interestingly, expression of *Txn1* was upregulated upon TLR-driven stimulation, potentially indicating an involvement of the Trx1 pathway in inflammatory responses (*Figure 4C*). Indeed, while WT BMDCs readily produced pro-inflammatory cytokines including IL-6, IL-12p40 and TNF-α in responses to each of the stimuli, this was strikingly impaired in *Txnrd1*-deficient BMDCs (*Figure 4D, E* and *Figure 4—figure supplement 1*), which was also evident at the transcriptional level (*Figure 4F,G*). Similarly, the secretion of IL-1β by TLR-primed BMDCs upon ATP, alum (Al(OH)$_3$) or nigericin stimulation was also defective in the absence of *Txnrd1* due to impaired transcription of the *Il1b* gene (*Figure 4H,I* and *Figure 4—figure supplement 2*). Together, these data suggest that the Trx1 system positively regulates the transcriptional program triggered by TLR ligands.

Considering the important role of IL-12 in the polarization of T helper type 1 (Th1) cells (*Langenkamp et al., 2000*), we next wondered whether reduced IL-12 production by *Txnrd1*-deficient BMDCs would affect their ability to drive Th1 polarization of CD4$^+$ T cells. Indeed, we measured a lower frequency of IFN-γ-producing CD4$^+$ T cells when naïve CD4$^+$ T cells were co-cultured with *Txnrd1*-deficient BMDCs (*Figure 4J*) compared to when they were co-cultured with WT controls. Therefore, the absence of the Trx1 system in BMDCs inhibits their capacity to induce pro-inflammatory, IFN-γ-producing Th1 cells.

## The Trx1 system regulates the DNA-binding activity of NF-κB to its response element

We next aimed to elucidate the mechanism of impaired pro-inflammatory responses in *Txnrd1*-deficient BMDCs. Upon binding of microbial ligands, TLR associate with adaptor molecules such as MyD88 to activate the IKK complex and mitogen-activated protein kinases, such as extracellular receptor kinase (Erk), p38 and c-Jun N-terminal kinase (JNK), which in turn influence the activity of the transcription factors NF-κB, AP1 and CREB, thereby ultimately regulating expression of pro-inflammatory cytokines (*Akira and Takeda, 2004*). Interestingly, we observed intact TLR signaling in *Txnrd1*-deficient BMDCs upon stimulation with LPS or R837, as assessed by the phosphorylation and consequent degradation of IκB-α and phosphorylation of Erk1/2 (*Figure 5A* and *Figure 5—figure supplement 1*), thus supporting the idea that early downstream events upon TLR stimulation are not affected by the absence of the Trx1 system.

Proteolysis of IκB-α results in the release and translocation of NF-κB p65 and p50 to the nucleus, where it induces the transcription of genes encoding pro-inflammatory cytokines (*Akira and Takeda, 2004*). Since no degradation defects of IκB-α were observed in the absence of *Txnrd1*, we next analyzed the translocation of NF-κB into the nucleus by microscopy. However, we observed no significant defects in the nuclear translocation of NF-κB p65 between *Txnrd1*-deficient and sufficient BMDCs (*Figure 5B,C*). Apart from this, we noticed the presence of cytosolic p65 punctate in the absence of the Trx1 system (*Figure 5—figure supplement 2*), but the relevance and the potential cellular consequences of this pattern are unknown at present and need further investigations. Overall, the microscopy analysis demonstrates that *Txnrd1* deficiency does not affect the nuclear translocation of NF-κB p65, therefore suggesting a downstream defect.

We next hypothesized that *Txnrd1* deficiency might affect the binding of NF-κB p65 to its DNA response element. Therefore, we stimulated *Txnrd1*-deficient and sufficient BMDCs with LPS and performed NF-κB p65 chromatin immunoprecipitation (ChIP) followed by RT-PCR to investigate p65 DNA binding to the promoters of inflammatory genes. In keeping with our hypothesis, we observed reduced NF-κB p65 DNA binding to the promoters of the pro-inflammatory genes *Il12b*, *Il1b* and *Il6*, and of the notorious NF-κB target gene *Nfkbia* (encoding IκB-α) (*Figure 5D*). Furthermore, we

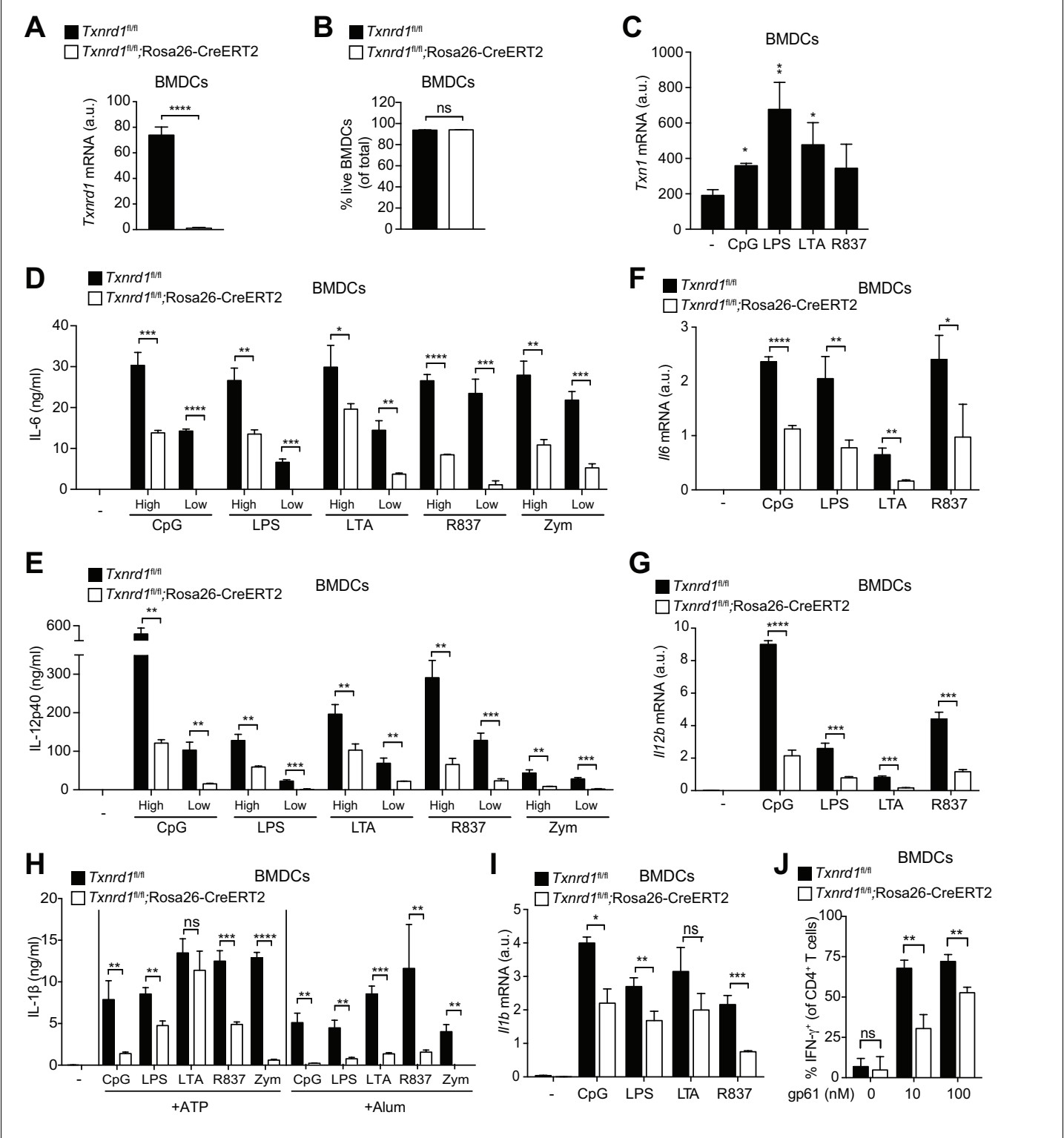

**Figure 4.** *Txnrd1*-deficient BMDCs undergo defective transcription of pro-inflammatory cytokines. (A–J) *Txnrd1*<sup>fl/fl</sup>;Rosa26-CreERT2 mice and *Txnrd1*<sup>fl/fl</sup> littermates were injected with TAM to delete the *Txnrd1* gene. Bone marrow cells were differentiated with GM-CSF to obtain BMDCs. (A) Analysis of *Txnrd1* mRNA by RT-PCR in cultured *Txnrd1*<sup>fl/fl</sup>;Rosa26-CreERT2 and *Txnrd1*<sup>fl/fl</sup> BMDCs for confirmation of gene deletion (n = 3). (B) Viability (eFluor780⁻Annexin-V⁻) of BMDCs was assessed via flow cytometry (n = 3). (C) BMDCs were primed with CpG (100 nM), LPS (100 ng/ml), LTA (1 μg/ml), R837 (5 μg/ml) for 7 hr, and expression of *Txn1* at the mRNA level was determined via RT-PCR (n = 3). (D, E) BMDCs were stimulated for 7 hr with CpG (100, 20 nM), LPS (100, 20 ng/ml), LTA (5, 1 μg/ml), R837 (5, 1 μg/ml), or zymosan (10, 2 μg/ml). 'High' and 'Low' indicate the concentration of the

*Figure 4 continued on next page*

Figure 4 continued

utilized stimulus. IL-6 (D) and IL-12p40 (E) were measured in supernatants by ELISA (n = 3). (F, G) BMDCs were stimulated for 7 hr with CpG (100 nM), LPS (100 ng/ml), LTA (1 µg/ml) or R837 (5 µg/ml), and the expression of *Il6* (F) and *Il12b* (G) was determined by RT-PCR (n = 3). (H) BMDCs were stimulated with CpG (100 nM), LPS (400 ng/ml), LTA (5 µg/ml), R837 (5 µg/ml) or zymosan (10 µg/ml) before the addition of 2 mM ATP or 200 µg/ml alum for 1 hr or 4 hr, respectively. The concentration of IL-1β in supernatants was determined by ELISA (n = 3). (I) BMDCs were stimulated with TLR ligands as in F,G), and expression of *Il1b* at the mRNA level was determined via RT-PCR (n = 3). (J) Naive, splenic, Smarta-1 transgenic CD4$^+$ T cells were co-cultured with *Txnrd1*-deficient BMDCs (or *Txnrd1*-sufficient BMDCs as a control) and the indicated concentrations of the GP$_{61-80}$ peptide. Shown are the frequencies of CD4$^+$ T cells producing IFN-γ$^+$ after restimulation with PMA/ionomycin (n = 3). Bar graphs represent mean + standard deviation. Data are representative of two (A–C, F, G, I, J) or four (D, E, H) independent experiments. For each panel, a representative experiment with replicates of in vitro culture conditions is shown. Student's t test (two-tailed, unpaired) was used to compare *Txnrd1*$^{fl/fl}$;Rosa26-CreERT2 and control *Txnrd1*$^{fl/fl}$ groups in (A, B, D–J): *p≤0.05; **p≤0.01; ***p≤0.001; ****p≤0.0001; ns, not significant. One-way ANOVA followed by Dunnett's correction was used in C (comparison to the unstimulated control): *p≤0.0332; **p≤0.0021.

The online version of this article includes the following figure supplement(s) for figure 4:

**Figure supplement 1.** *Txnrd1*-deficient BMDCs display an impaired secretion of TNF-α.

**Figure supplement 2.** *Txnrd1*-deficient BMDCs display an impaired secretion of IL-1β.

validated our ChIP results using an ELISA-based method for detecting specific p65 transcription factor DNA binding in nuclear extracts (*Figure 5E*). Thus, these data indicate that the Trx1 system positively regulates transcription of pro-inflammatory cytokines by promoting the binding of NF-κB to its DNA response element in BMDCs.

As the Trx1 system is an important player in maintaining cellular redox homeostasis (*Mustacich and Powis, 2000*), we next investigated whether deletion of *Txnrd1* would impact ROS levels and consequently cytokine secretion in BMDCs. Despite a significant increase in total cellular ROS levels (*Figure 5—figure supplement 3A*), the reduction in IL-12p40 secretion by *Txnrd1*-deficient BMDCs was not restored by the supplementation of a panel of antioxidants (*Figure 5—figure supplement 3B,C*). Importantly, we verified as a control that the antioxidant catalase scavenged excessive ROS in the absence of the *Txnrd1* (*Figure 5—figure supplement 3D*). Together, these results suggest that the defect in pro-inflammatory cytokine secretion is not a consequence of a general increase of ROS but is due to the compromised Trx1-mediated regulation of the redox status of NF-κB, which interferes with NF-κB p65 binding to the DNA.

## The Trx1 system promotes M1 macrophage polarization

The BMDC culture is known to comprise both conventional DCs and monocyte-derived macrophages (*Helft et al., 2015*) with the latter only possessing the capacity of inflammasome activation and IL-1β release (*Erlich et al., 2019*). To address the role of the Trx1 system in pro-inflammatory responses of macrophages, we harvested bone marrow from TAM-treated *Txnrd1*$^{fl/fl}$;Rosa26-CreERT2 and *Txnrd1*$^{fl/fl}$ littermate controls and generated BMDMs in culture with macrophage colony-stimulating factor (M-CSF). We first verified how the deletion of *Txnrd1* would affect classical (M1) and alternative (M2) macrophage activation (*Figure 6A*), which are known to mediate inflammatory responses and tissue repair, respectively. Interestingly, we observed increased expression of *Txnrd1* and *Txn1* in M1 compared to M2 macrophages and in BMDMs stimulated with a variety of TLR ligands (*Figure 6B,C* and *Figure 6—figure supplement 1A*), consistent with a possible role in maintaining redox homeostasis in ROS-generating M1 macrophages; however, no difference in expression of the Trx1 system inhibitor *Txnip* was detected (*Figure 6—figure supplement 1B*). In keeping with the induction of the Trx1 system in M1 macrophages, we observed lower expression of the M1 markers *Nos2*, *Cd38*, *Hif1a* and *Gpr18* in M1-polarized *Txnrd1*-deficient cells compared to the WT controls (*Figure 6D* and *Figure 6—figure supplement 1C–E*). A similar impairment in Nos2 expression was additionally confirmed at the protein level by flow cytometry (*Figure 6E*). However, *Txnrd1* deletion did not affect M2 polarization as measured by the RNA expression of the known M2 markers *Fizz1*, *Arg1* and *Ym1* (*Figure 6F* and *Figure 6—figure supplement 1F,G*), and protein levels of Relmα, CD301b, PDL2 and CD206 (*Figure 6G* and *Figure 6—figure supplement 1H–J*). Importantly, we excluded that defective M1 polarization was a consequence of cell death, since *Txnrd1*-deficient and sufficient BMDMs underwent cell death to a similar extent (*Figure 6H* and *Figure 6—figure supplement 1K*). Taken together, these results demonstrate that the Trx1 system is induced by LPS/IFN-γ stimulation and promotes a M1-macrophage phenotype.

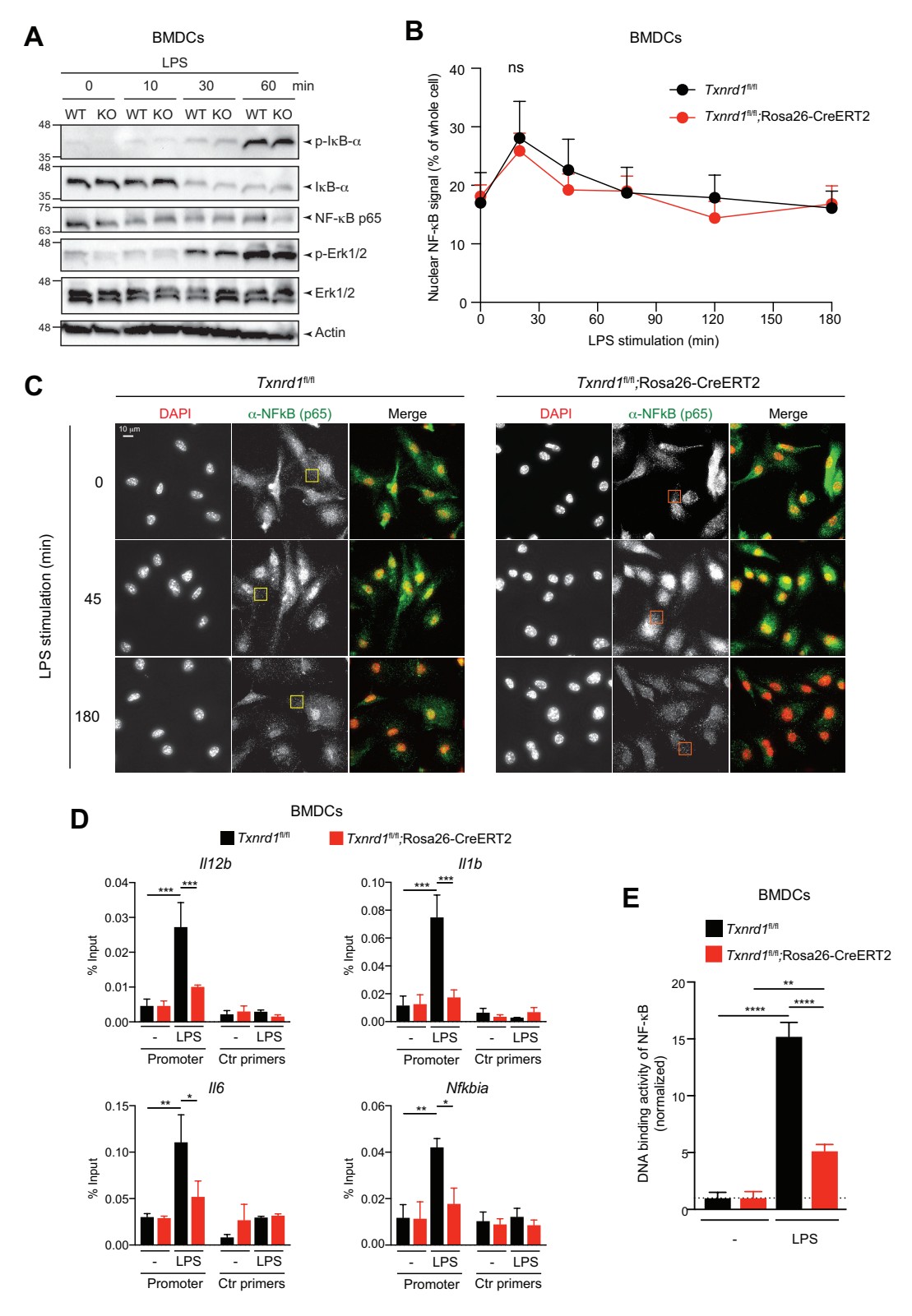

**Figure 5.** The Trx1 system positively regulates the binding activity of NF-κB p65 to the DNA in BMDCs. (A–E) *Txnrd1*<sup>fl/fl</sup>;Rosa26-CreERT2 mice and *Txnrd1*<sup>fl/fl</sup> littermates were injected with TAM to delete the *Txnrd1* gene, and bone marrow cells were differentiated with GM-CSF to obtain BMDCs. (A) BMDCs were stimulated with LPS (400 ng/ml) for 10, 30, or 60 min and lysed for western blot. Expression of phospho-IκB-α, respective IκB-α, NF-κB p65, phospho-Erk1/2 and respective Erk1/2 was assessed with β-actin as a loading control. (B, C) WT or *Txnrd1*-deficient BMDCs were fixed at the
*Figure 5 continued on next page*

*Figure 5 continued*

indicated time points post LPS treatment, stained for DNA (DAPI), NF-κB p65 and actin (Phalloidin), and imaged using a DeltaVision system. Approximately 10 randomly chosen imaging fields encapturing a total of 50–100 nuclei were analyzed per sample per condition. (B) Nuclear and whole-cell masks were made using the DAPI and phalloidin channels, and NF-κB signal intensity within the masks was quantified. Shown is the nuclear NF-κB signal strength plotted as percentage of whole-cell NF-κB signal. (C) Depicted are example images of the samples of indicated times points post LPS treatment. In the merged images, DAPI and anti-NF-κB channels are shown in red and green, respectively. Scale bar represents 10 μm (top-left panel). Squares indicate fields, which are magnified in *Figure 5—figure supplement 2*. (D) WT or *Txnrd1*-deficient BMDCs were stimulated with LPS (400 ng/ml) for 100 min, and the recruitment of NF-κB p65 to the *Il12b* (top-left), *Il1b* (top-right), *Il6* (bottom-left) and *Nfkbia* (bottom-right) promoters was assessed by p65 chromatin immunoprecipitation (ChIP) analysis and quantified by RT-PCR. 'Promoter' indicates the utilization of a primer pairs that amplify a fragment close to the NF-κB binding sites at the promoter region of the indicated genes, whereas 'Ctr primers' indicate primer pairs that were used as a control to amplify a region several kilobases away from the NF-κB binding sites (n = 2). (E) The NF-κB p65 binding activity to its DNA response element in nuclear extracts from BMDCs stimulated with LPS (400 ng/ml) for 40 min was assessed by an ELISA-based method (n = 3). Bar graphs and dot plots represent mean + standard deviation. Data are representative of two (A–D) or three (E) independent experiments. For each panel, a representative experiment with technical replicates is shown (D, E). Two-way ANOVA adjusted by Bonferroni's multiple comparison test was used in B: ns, not significant. One-way ANOVA adjusted by Tukey's multiple comparison test was used in D, E: *p≤0.0332; **p≤0.0021; ***p≤0.0002; ****p≤0.0001.

The online version of this article includes the following figure supplement(s) for figure 5:

**Figure supplement 1.** *Txnrd1* deficiency does not affect TLR signaling.
**Figure supplement 2.** Magnification of the microscopy images of *Figure 5C*.
**Figure supplement 3.** Antioxidant supplementation does not rescue IL-12p40 cytokine production in *Txnrd1*-deficient BMDCs.

## The GSH/Grx system can promote the DNA-binding activity of NF-κB in *Txnrd1*-deficient BMDMs

Since M1 macrophages secrete pro-inflammatory cytokines during immune responses, we next stimulated *Txnrd1*-deficient and sufficient BMDMs with a variety of TLR ligands. To our surprise and in striking contrast to the results obtained with BMDCs, IL-12p40 levels remained unaffected (*Figure 6I*), while production of IL-6 (*Figure 6J*) and TNF-α (*Figure 6—figure supplement 2*) was even increased in *Txnrd1*-deficient BMDMs compared to controls. Importantly, *Txnrd1* gene deletion in knockout cells was confirmed at the mRNA level (*Figure 6K*). Along the same lines, *Txnrd1^fl/fl*; Rosa26-CreERT2 BMDMs displayed significantly higher *Il12b* and *Il6* expression at the transcriptional level compared to *Txnrd1^fl/fl* control BMDMs (*Figure 6L,M*) upon stimulation, which may suggest the presence of a compensatory mechanism.

Overlapping activities of the Trx and GSH/Grx systems have been suggested in different biological responses, but their relative contribution in a particular response in distinct cell types is barely understood (*Lillig and Holmgren, 2007*). Interestingly, we observed that all Grx isoforms were higher expressed in BMDM compared to BMDC cultures (*Figure 6N*). Considering the fact that *Txnrd1* is critical for the transcription of pro-inflammatory cytokines in BMDCs but not in BMDMs as shown above, we next wondered whether the alternative GSH/Grx system compensates for the absence of the Trx1 system in BMDMs. To address this, we treated *Txnrd1^fl/fl*;Rosa26-CreERT2 and *Txnrd1^fl/fl* BMDMs with BSO prior to stimulation with TLR ligands to deplete GSH and thereby interfere with Grx functions. Treatment of *Txnrd1^fl/fl*;Rosa26-CreERT2 BMDMs with BSO drastically reduced IL-12p40 secretion to minimal levels (*Figure 6O* and *Figure 6—figure supplement 3A*), without affecting viability (*Figure 6—figure supplement 3B*). In addition, BSO-treatment did not affect cytokine secretion by WT BMDMs (*Figure 6O* and *Figure 6—figure supplement 3A*), suggesting that the Trx1 system alone is sufficient for sustaining cytokine production.

We next investigated how *Txnrd1* deficiency influences DNA binding activity of NF-κB to its response element in BMDMs. In keeping with no decrease in IL-6, IL-12p40 and TNF-α secretion in the absence of the Trx1 system, we observed that NF-κB p65 DNA binding activity was not affected by *Txnrd1* deletion (*Figure 6P*). Interestingly, however, blocking the GSH/Grx pathway by BSO treatment in *Txnrd1*-deficient BMDMs turned out to completely abrogate the DNA-binding capacity of NF-κB p65, thereby confirming the compensatory activity of the GSH/Grx system in BMDMs (*Figure 6P*). Overall, here we show that while BMDCs are critically dependent on the Trx1 system for NF-κB-mediated pro-inflammatory cytokine secretion, BMDMs can utilize both the Trx1 and GSH/Grx systems to fuel redox reactions aimed at promoting NF-κB-binding activity and consequent transcription of pro-inflammatory target genes.

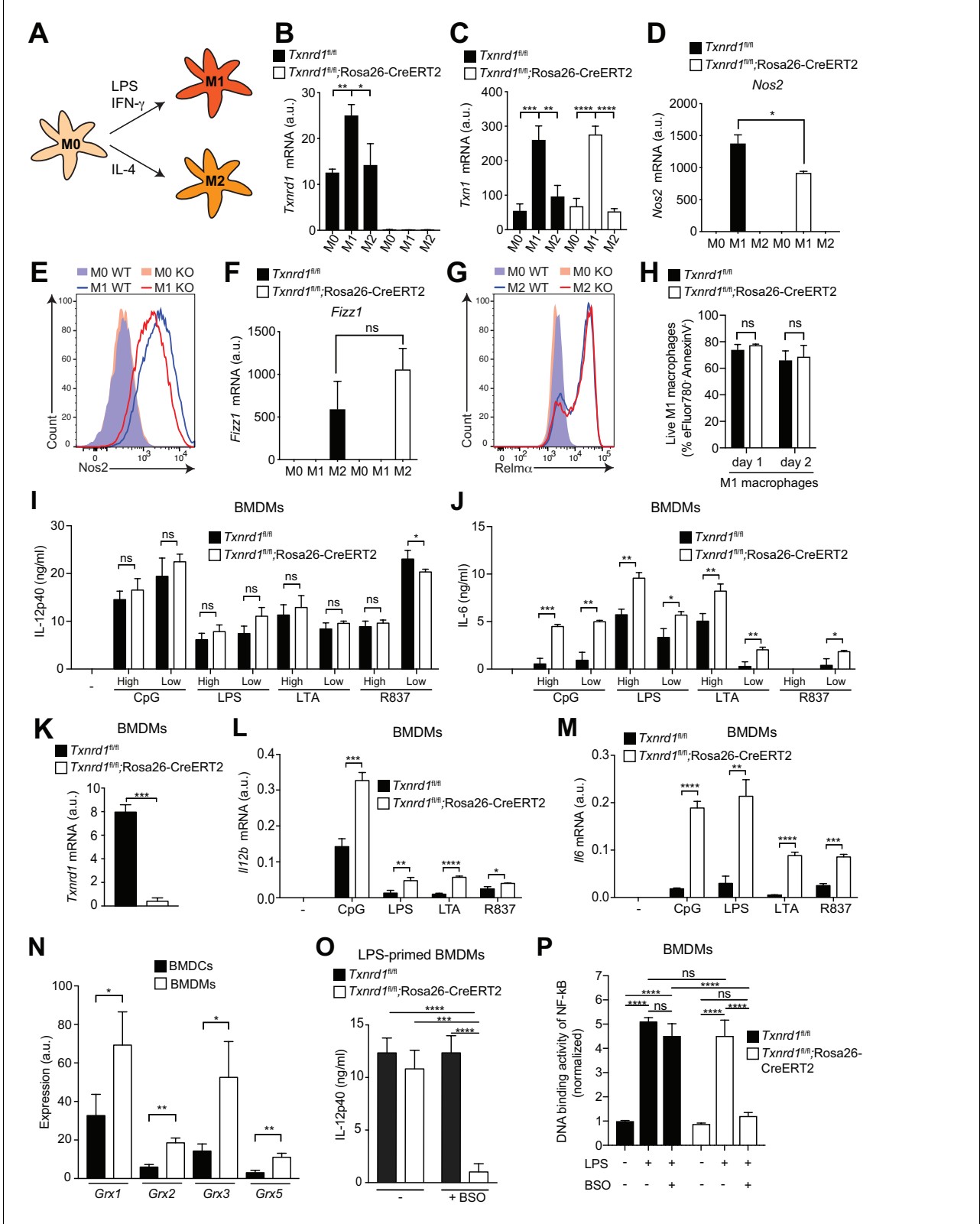

**Figure 6.** The Trx1 system is critical for proper M1 polarization but dispensable for promoting NF-κB binding to target DNA in BMDMs. (A–P) Bone marrows from *Txnrd1^{fl/fl}*;Rosa26-CreERT2 mice and control *Txnrd1^{fl/fl}* littermates treated with TAM were differentiated with M-CSF to obtain BMDMs. (A–H) BMDMs were polarized toward M1 and M2 with LPS/IFN-γ and IL-4, respectively, or left untreated (M0). (A) Schematic showing the polarization assay. (B, C) Shown are the expression levels of *Txnrd1* (B) and *Txn1* (C) at the mRNA level determined by RT-PCR (n = 3). (D, E) Depicted is the level of
*Figure 6 continued on next page*

*Figure 6 continued*

the M1 marker Nos2 at the mRNA level determined by RT-PCR (D); n = 3) and at the protein level assessed by flow cytometry (E). (F, G) Shown is the level of the M2 marker Relmα at the mRNA level (*Fizz1*) determined by RT-PCR (F); n = 3) and at the protein level assessed by flow cytometry (G). (H) Cell survival of M1 macrophages (eFluor780⁻Annexin-V⁻) polarized either for 1 or 2 days with LPS/IFN-γ was assessed via flow cytometry (n = 3). (I, J) BMDMs were stimulated for 7 hr with CpG (400, 100 nM), LPS (100, 20 ng/ml), LTA (5, 1 µg/ml), or R837 (5, 1 µg/ml), and IL-12p40 (I) and IL-6 (J) were measured in supernatants by ELISA. 'High' and 'Low' indicate the concentration of the utilized stimulus (n = 3). (K) Analysis of *Txnrd1* mRNA in BMDMs assessed by RT-PCR for confirmation of gene deletion (n = 3). (L, M) Shown is the mRNA expression of *Il12b* (L) and *Il6* (M) determined via RT-PCR in BMDMs stimulated for 7 hr with CpG (100 nM), LPS (100 ng/ml), LTA (1 µg/ml) or R837 (5 µg/ml; n = 3). (N) Shown are the expression levels of the indicated Grx isoforms in WT BMDCs and BMDMs (n = 3). (O) BMDMs were treated overnight with BSO (3 µM) before stimulation with LPS (400 ng/ml) for 7 hr. The concentration of IL-12p40 in supernatants was subsequently determined by ELISA (n = 3). (P) Shown is the DNA binding activity of NF-κB p65 in nuclear extracts from *Txnrd1^fl/fl*;Rosa26-CreERT2 and control *Txnrd1^fl/fl* BMDMs treated overnight with 3 µM BSO (or medium as a control) before stimulation with LPS (400 ng/ml) for 90 min (n = 3). Bar graphs represent mean + standard deviation. Data are representative of two (B–H, K–P) or three (I, J) independent experiments. For each panel, a representative experiment with replicates of in vitro culture conditions is shown (B–D, F, H–P). One-way ANOVA adjusted by Tukey's multiple comparison test was used in B-D, F, O, P: *p≤0.0332; **p≤0.0021; ***p≤0.0002; ****p≤0.0001; ns, not significant. Student's *t* test (two-tailed, unpaired) was used for two-group analysis in H–N: *p≤0.05; **p≤0.01; ***p≤0.001; ****p≤0.0001; ns, not significant.

The online version of this article includes the following figure supplement(s) for figure 6:

**Figure supplement 1.** The Trx1 system promotes M1 but not M2 macrophage polarization.

**Figure supplement 2.** *Txnrd1* deficiency does not impair TNF-α production.

**Figure supplement 3.** The GSH/Grx system compensates for the absence of the Trx1 pathway during IL-12p40 production by *Txnrd1*-deficient BMDMs.

## The Trx1 system but not Txnip positively regulates NLRP3 inflammasome-driven IL-1β maturation

An intact signal one in *Txnrd1*-deficient BMDMs, in contrast to *Txnrd1*-deficient BMDCs, allowed us to study the quality of signal two and inflammasome activation in the absence of *Txnrd1*. Interestingly, release of mature IL-1β induced by the NLRP3 activators ATP and alum following priming with various TLR ligands was abrogated in *Txnrd1*-deficient BMDMs (**Figure 7A** and **Figure 7—figure supplement 1A**) as shown using an ELISA assay known to display a much higher sensitivity for mature IL-1β than for its pro-form (**Dick et al., 2016**). This was additionally confirmed by western blotting showing strikingly reduced release of processed IL-1β upon NLRP3 inflammasome activation, while levels of pro-IL-1β in cell lysates (**Figure 7B** and **Figure 7—figure supplement 1B**) and transcription of *Il1b* mRNA were unaffected in *Txnrd1*-deficient BMDMs upon TLR priming (**Figure 7C**). Consistent with a defective NLRP3 inflammasome, the processing of caspase-1 to its active form was also dramatically impaired (**Figure 7B** and **Figure 7—figure supplement 1B**). Overall, these data demonstrate that the Trx1 system promotes NLRP3 inflammasome activation and consequently secretion of the mature IL-1β.

Intraperitoneal administration of monosodium urate (MSU) is well known to elicit NLRP3-dependent production of mature IL-1β in mouse peritoneum leading to a massive infiltration of neutrophils (**Martinon et al., 2006**). In line with the in vitro defects, we found that *Txnrd1*-deficient mice display an impaired neutrophil influx (**Figure 7D**) and reduced IL-1β production (**Figure 7E**) upon MSU injection compared to WT mice. Therefore, these results further confirm that the Trx1 system plays a critical role in NLRP3 inflammasome activation and IL-1β secretion in vivo.

The Trx1 system is a major player in the control of cellular antioxidant responses (**Mustacich and Powis, 2000**). Therefore, we next hypothesized that *Txnrd1* deficiency may influence ROS homeostasis and consequently negatively affect the normal processing of IL-1β by the NLRP3 inflammasome. To test this possibility, we first quantified cellular ROS in the absence of the Trx1 system by CM-H₂DCFDA staining and observed that *Txnrd1*-deficient BMDMs accumulated ROS to a higher extent compared to control cells after treatment with LPS or R837 (**Figure 7F,G**). Notably, we additionally found that supplementation with a cell-permeable version of catalase, one of the main cellular antioxidants involved in the neutralization of hydrogen peroxide, can restore ATP- and alum-induced IL-1β production in *Txnrd1*-deficient BMDMs up to WT levels (**Figure 7H** and **Figure 7—figure supplement 2**). As a control, we verified that addition of catalase does not affect the expression of the *Il1b* gene (**Figure 7—figure supplement 3**). By contrast, other classical antioxidants, such as N-acetyl-L-cysteine (NAC), ascorbic acid (AA), dithiothreitol (DTT), and diphenyleneiodonium (DPI), did not rescue the impaired IL-1β production in the absence of the Trx1 system (**Figure 7—figure**

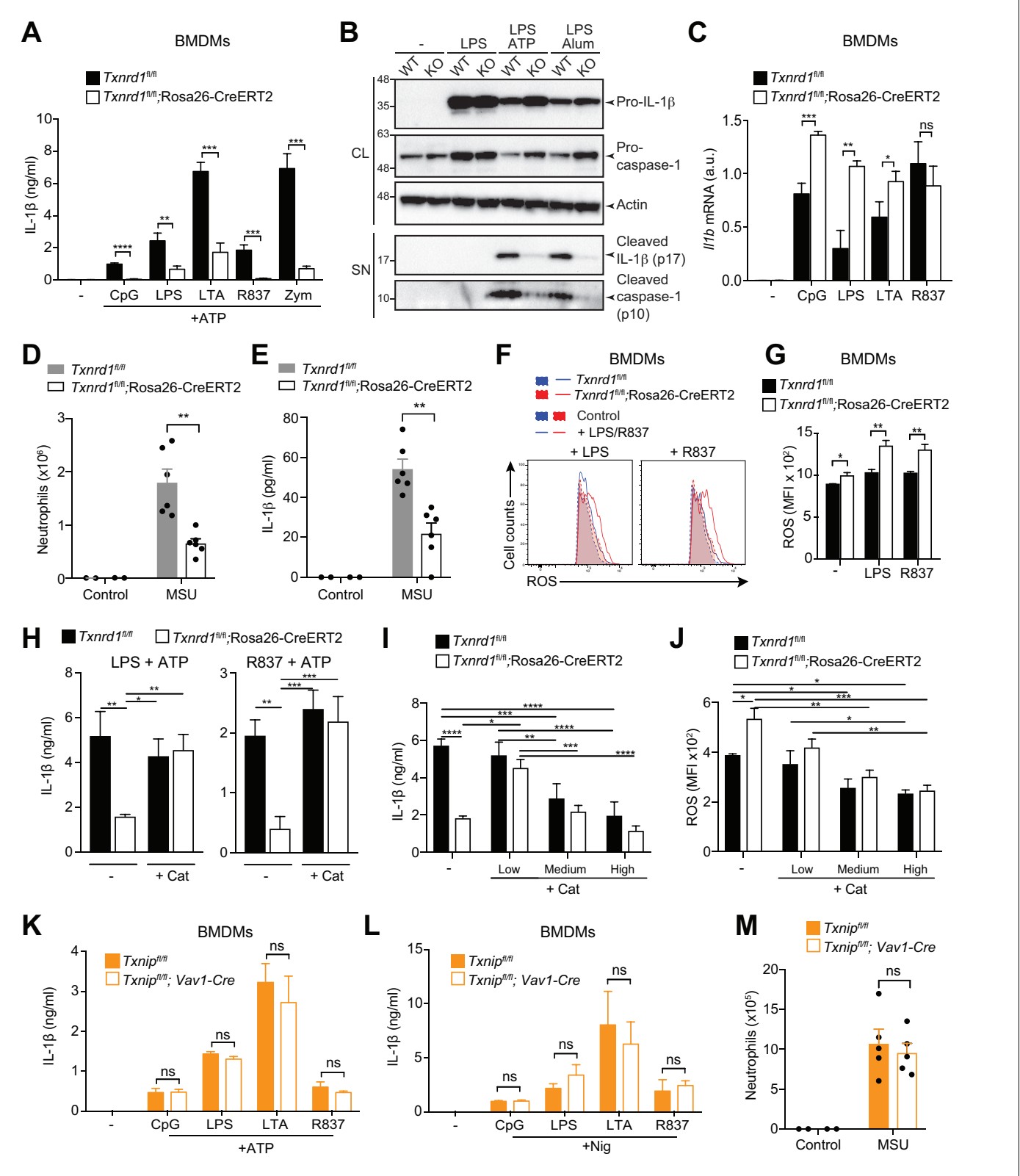

**Figure 7.** Trx1 but not Txnip promotes NLRP3-dependent IL-1β responses. (A–C, F–L) Bone marrows from the indicated genotypes (from TAM-treated *Txnrd1^{fl/fl}*;Rosa26-CreERT2 and control *Txnrd1^{fl/fl}* littermates or from *Txnip^{fl/fl}*;Vav1-Cre and *Txnip^{fl/fl}* control mice) were differentiated with M-CSF to obtain BMDMs. (A) *Txnrd1*-deficient and control BMDMs were primed with CpG (100 nM), LPS (400 ng/ml), LTA (5 μg/ml), R837 (5 μg/ml) or zymosan (10 μg/ml) before stimulation with 2 mM ATP for 1 hr. The concentration of IL-1β in supernatants was then determined by ELISA (n = 3). (B) *Txnrd1^{fl/fl}*

*Figure 7 continued on next page*

*Figure 7 continued*

(WT) *and Txnrd1^{fl/fl}*;Rosa26-CreERT2 (KO) BMDMs were primed with LPS (400 ng/ml) before stimulation with 2 mM ATP or 200 µg/ml alum. Levels and processing of IL-1β and caspase-1 in the cell lysate (CL) and supernatant (SN) were assessed by western blot with β-actin as a loading control. (C) *Txnrd1*-deficient and control BMDMs were stimulated with CpG (100 nM), LPS (100 ng/ml), LTA (1 µg/ml), R837 (5 µg/ml) for 7 hr, and expression of *Il1b* at the mRNA level was determined via RT-PCR (n = 3). (D, E) TAM-treated *Txnrd1^{fl/fl}*;Rosa26-CreERT2 and control *Txnrd1^{fl/fl}* littermates were intraperitoneally injected with MSU crystals, and peritoneal lavage fluid was collected 8 hr later. Depicted are the total counts of infiltrating neutrophils (D) and the IL-1β levels measured by ELISA (E); n = 2 mice for controls, and n = 6 mice for MSU-treated groups). (F, G) Quantification of ROS levels in unstimulated BMDMs as control or after 4 hr of stimulation with LPS (400 ng/ml) or R837 (5 µg/ml) stained with CM-H$_2$DCFDA via flow cytometry. Representative FACS plots (F) and mean fluorescence intensities (MFI; G) are shown (n = 3). (H) BMDMs were treated overnight with the antioxidant catalase-polyethylene glycol (Cat; 62.5 U/ml) before priming with LPS (400 ng/ml; left) or R837 (5 µg/ml; right) followed by stimulation with ATP. The secreted IL-1β was determined by ELISA (n = 3). (I, J) BMDMs were treated overnight with the antioxidant catalase-polyethylene glycol (Cat; 'low' is 62.5 U/ml; 'medium' is 250 U/ml; 'high' is 1000 U/ml) before stimulation (n = 3). (I) IL-1β secretion after priming with LPS (400 ng/ml) and stimulation with ATP was determined by ELISA. (J) Quantification of ROS levels after 4 hr of stimulation with LPS (400 ng/ml) was performed by flow cytometric staining with CM-H$_2$DCFDA. (K, L) *Txnip*-deficient and control BMDMs were primed with CpG (100 nM), LPS (400 ng/ml), LTA (5 µg/ml), or R837 (5 µg/ml) before stimulation with 2 mM ATP (K) or 15 µM nigericin (Nig; L) for 1 hr. The concentration of IL-1β in supernatants was determined by ELISA (n = 3). (M) *Txnip^{fl/fl}*;Vav1-Cre and *Txnip^{fl/fl}* control mice were intraperitoneally injected with MSU crystals, and peritoneal lavage fluid was collected 8 hr later. Shown are the total counts of infiltrating neutrophils (n = 2 mice for controls, and n = 5 mice for MSU-treated groups). Bar graphs represent mean + standard deviation, and dots in the plots indicate individual mice. Data are representative of four (A), two (B–J, M), or three (K, L) independent experiments. For each panel, a representative experiment with either biological replicates (D, E, M) or replicates of in vitro culture conditions (A, C, G–L) is shown. Student's t test (two-tailed, unpaired) was used for two-groups analysis (A, C, G, K, L): *p≤0.05; **p≤0.01; ***p≤0.001; ****p≤0.0001; ns, not significant. One-way ANOVA adjusted by Tukey's multiple comparison test was used in D, E, H-J, M: *p≤0.0332; **p≤0.0021; ***p≤0.0002; ****p≤0.0001. Legends for Figure supplements.

The online version of this article includes the following figure supplement(s) for figure 7:

**Figure supplement 1.** The Trx1 system is required for NLRP3 inflammasome responses.

**Figure supplement 2.** Catalase treatment restores IL-1β production in the absence of the Trx1 system.

**Figure supplement 3.** Catalase treatment does not affect transcriptional IL-1β levels.

**Figure supplement 4.** Supplementation with the indicated antioxidants does not restore IL-1β production in the absence of the Trx1 system in BMDMs.

**Figure supplement 5.** *Txnip* deficiency does not affect cytokine secretion in BMDMs.

supplement 4). As ROS are commonly described as inducers of the NLRP3 inflammasome (*Tschopp and Schroder, 2010*), we next aimed at better understanding why the antioxidant catalase restores IL-1β production in the absence of the Trx1 system. Therefore, we incubated *Txnrd1*-deficient and -sufficient BMDMs with increasing concentrations of catalase and measured IL-1β secretion and ROS levels upon LPS and ATP stimulation. We observed that only low concentrations of catalase restored defective IL-1β production in *Txnrd1*-deficient cells (*Figure 7I*) and concomitantly reduced ROS to levels observed in WT cells (*Figure 7J*). However, higher concentrations of the antioxidant catalase (described as 'medium' and 'high' in the graphs) more potently inhibited cellular ROS and consequently IL-1β secretion upon NLRP3 inflammasome activation in both WT and *Txnrd1*-deficient BMDMs (*Figure 7I,J*), in line with the general view of ROS as triggers of the NLRP3 inflammasome (*Tschopp and Schroder, 2010*). Altogether, these results demonstrate that the Trx1 pathway is crucial for IL-1β production in macrophages by prevention of an excessive accumulation of hydrogen peroxide.

Our data suggest that excessive levels of ROS impair inflammasome activation. Contrary to these observations, the Trx1 inhibitor Txnip has been described as a redox-sensitive ligand of the NLRP3 inflammasome that promotes mature IL-1β production in response to ROS (*Zhou et al., 2010*). Therefore, to reconcile this and comprehensively understand how the Trx1-Txnip system regulates IL-1β release, we generated *Txnip^{fl/fl}*;Vav1-Cre mice by crossing mice with *loxP*-flanked *Txnip* alleles to mice expressing Cre recombinase from the *Vav1* promoter, and differentiated in vitro BMDMs as described above. Although *Txnip* deletion was complete at the mRNA level (*Figure 7—figure supplement 5A*), *Txnip*-deficient BMDMs did not show any defects in IL-12p40 (*Figure 7—figure supplement 5B*) and IL-1β secretion (*Figure 7K,L* and *Figure 7—figure supplement 5C*) compared to WT controls. Moreover, administration of MSU into *Txnip*-deficient and sufficient mice elicited a comparable influx of neutrophils in the peritoneum (*Figure 7M*), thereby further supporting the dispensability of Txnip in NLRP3 inflammasome activation. Together, these data demonstrate that Trx1 but not Txnip is critical for NLRP3-dependent processing of IL-1β.

## Discussion

In this study, we assessed the role of the Trx1 system in the development, homeostatic maintenance, and inflammatory responses of myeloid cells. We found that the Trx1 pathway is not required for development and maintenance of tissue and blood monocytes, DCs, neutrophils, eosinophils and macrophages due to their flexibility in rearranging their redox system toward the GSH/Grx pathway to fuel cellular thiol-based reactions when the Trx1 system is compromised, in contrast to its critical role for the development of T cells (*Muri et al., 2018*). However, in a situation of emergency myelopoiesis driven by endotoxin, when proliferation of myeloid precursors bursts (*Boettcher et al., 2012*), the GSH/Grx pathway was insufficient to fully compensate for the absence of the Trx1 system and fewer neutrophils were found in blood of TAM-treated *Txnrd1^{fl/fl}*;Rosa26-CreERT2 mice. This may be either due to less efficient donation of electrons to RNR by the GSH/Grx system at the last step of the PPP or due to a delay in mobilization of this backup system. Indeed, we previously reported a delay in engagement of the Grx pathway in germinal center responses of follicular B cells lacking *Txnrd1* (*Muri et al., 2019b*).

Importantly, we have established a critical role of the Trx1 system for pro-inflammatory responses in both DCs and macrophages. Using BMDCs, we showed that Trx1 provides reducing equivalents for the redox-dependent binding of the NF-κB p65 transcription factor to target DNA in DCs. The transcription factor NF-κB serves as a crucial mediator of inflammatory responses by inducing the expression of numerous pro-inflammatory cytokine and chemokine genes among others (*Liu et al., 2017*). NF-κB regulation is tightly controlled both in the cytoplasm and in the nucleus, and many factors and conditions are known to control NF-κB induction including ROS (*Gloire and Piette, 2009*; *Kabe et al., 2005*; *Schreck et al., 1991*). Interestingly, while redox regulation of NF-κB activity by the Trx1 antioxidant pathway has been proposed, its contribution in inflammation remains controversial because both activating and inhibitory effects have been described (*Isakov et al., 2014*; *Kabe et al., 2005*; *Lillig and Holmgren, 2007*; *Morgan and Liu, 2011*). Our results demonstrate that the Trx1 system promotes the DNA-binding capacity of NF-κB in the nucleus rather than its activation in the cytoplasm or nuclear translocation. Trx1 promotes NF-κB DNA binding probably by structural modification of a component(s) (e.g. via reduction of disulfide bonds) of the NF-κB binding complex rather than through indirect effects of ROS, since supplementation with different antioxidants did not restore NF-κB activity in *Txnrd1*-deficient BMDCs.

GM-CSF-driven BMDCs are known to comprise a mixture of conventional DCs, monocyte-derived DCs and macrophages (*Helft et al., 2015*), with the latter population only capable of inflammasome-dependent IL-1β production (*Erlich et al., 2019*). Therefore, we aimed to verify our results in BMDMs generated with M-CSF, which are thought to consist of a pure population of macrophages. However, in contrast to BMDCs, we surprisingly found that BMDMs displayed normal NF-κB DNA binding activity and transcription of pro-inflammatory genes in the absence of the Trx1 system and showed that this is due to higher expression of the cellular Grx proteins in BMDMs compared to BMDCs. Indeed, we showed that BMDMs but not BMDCs have the capability to engage the Grx pathway when the Trx1 system is genetically deleted. Thus, Trx1 activates NF-κB transcription, and depending on the expression levels of Grx proteins in different myeloid-cell types, *Txnrd1*-deficiency may have more or less pronounced outcomes. Myeloid cells are known to generate high amounts of ROS by NADPH oxidases to fight intracellular microbes, which may explain the requirement of robust ROS scavenging systems (*Lambeth, 2004*).

Inflammasomes are cytoplasmic multiprotein complexes that trigger IL-1β processing and release (*Broz and Dixit, 2016*). Although Txnip has been suggested as a redox-sensitive ligand of NLRP3 that links ROS generation to NLRP3 inflammasome activation (*Zhou et al., 2010*), we here showed that NLRP3-dependent IL-1β production in macrophages is independent of Txnip both in vitro and in vivo. In contrast, our study identifies the Txnip-interacting partner Trx1 as a critical regulator of inflammasome-dependent IL-1β release in macrophages. In the absence of the Trx1 system, we observed drastically impaired IL-1β production in response to classical NLRP3 stimuli in vitro and during MSU-induced peritonitis in vivo, while pro-*Il1b* transcription was unaltered. Interestingly, *Txnrd1*-deficient BMDMs displayed excessive intracellular ROS levels upon TLR triggering, and decomposition of hydrogen peroxide to water and oxygen with a cell-permeable form of the enzyme catalase restored IL-1β production. Notably, other anti-oxidants lacked this capacity, indicating that local peroxide detoxification by Trx1 is critical for NLRP3 inflammasome activation. Thus, our

findings suggest that moderate ROS levels act as triggers for the NLRP3 inflammasome in line with the known role of mitochondrial ROS released from stressed mitochondria (*Latz et al., 2013*; *Zhou et al., 2011*), while excessive ROS levels display an opposite effect and inhibit NLRP3 activation. Excessive ROS concentrations may lead to the oxidation of particular components of the inflammasome machinery (e.g. ASC, NLRP3, or caspase-1), with detrimental consequences for their ability to assemble and/or cleave IL-1β. Likely candidates might be oxidations on cysteine (Cys-disulfide; Cys-sulfenic acid; Cys-sulfinic acid; Cys-sulfonic acid) and on methionine (Met-sulfoxide), which are generally caused by oxidizing agents such as ROS. In keeping with this hypothesis, nitrogen oxide (NO) has been reported to induce NLRP3 and caspase-1 nitrosylation and thereby inhibit NLRP3 inflammasome activity (*Kim et al., 1998*; *Mishra et al., 2013*). Taken together, ROS must be kept within a restricted window of concentrations to allow inflammasome activation, where both an excess or an undersupply prevent it. A non-linear dose response to a stressor (e.g. ROS) with a certain effect at low doses and the reverse at high doses has been referred to as hormesis (*Ristow, 2014*; *Ristow and Schmeisser, 2011*). Our results are in line with an hormetic regulation of NLRP3 inflammasome activity by ROS, which promote its assembly at moderate concentrations, while they inhibit it at low and high concentrations.

While classically activated M1 macrophages are associated with inflammatory responses, alternatively activated M2 macrophages are described as anti-inflammatory, thereby promoting tissue remodeling and resolution of inflammation (*Chinetti-Gbaguidi et al., 2015*). Interestingly, extensive research has demonstrated that the distinct metabolic phenotypes of these two types of macrophages dictate their function. For instance, M1 macrophages undergo increased rates of glycolysis compared to M2 macrophages, allowing rapid ATP production and sustainment of the PPP in order to generate NADPH for ROS production and de novo nucleotide biosynthesis (*Kelly and O'Neill, 2015*; *O'Neill, 2015*; *Russell et al., 2019*). We found that LPS/IFN-γ but not IL-4 stimulation leads to an induction of the Trx1 system, which is in line with the observed impairment in M1 polarization in the absence of the Trx1 pathway. As discussed above, Trx1 donates electrons to RNR, which catalyzes the rate-limiting step of the PPP, leading to nucleotide biosynthesis (*Holmgren and Sengupta, 2010*). Therefore, the importance of the Trx1 system in M1 polarization is consistent with the high rates of the PPP in these cells. Why M1 macrophages require such enormous amounts of DNA building blocks despite their low proliferative capacity is unknown, although it has been suggested that they need them to generate different RNA populations, such as microRNAs and long non-coding RNAs (*O'Neill et al., 2016*). In *Txnrd1*-deficient M1 macrophages, reduced levels of these regulatory RNAs may impact proper M1 polarization. M2 macrophages, on the other hand, seem to suppress the PPP (*Haschemi et al., 2012*). Thus, this observation is fully in line with the dispensability of the Trx1 system in fueling the PPP during M2 polarization.

Our results show that the cellular redox systems of BMDCs and BMDMs differ considerably, with each providing valuable information to the mechanism. The defect of NF-κB mediated transcription (also referred to as signal 1 in the inflammasome field) in *Txnrd1*-deficient BMDCs did not allow to study inflammasome activity due to impaired transcription of pro-IL1β mRNA, which was intact in BMDMs and therefore allowing to unravel the function of Trx1 in NLRP3 activation. Importantly, we confirmed impaired NLRP3 inflammasome activity in vivo by treatment of *Txnrd1*-deficient mice with MSU. The differential redox regulation in BMDCs and BMDMs may reflect distinct requirements of the Trx1 and GSH/Grx pathways for the inflammatory response of DCs, tissue macrophages and monocyte-derived macrophages in vivo.

In conclusion, our study reveals a key role of the Trx1 system for NF-κB-mediated inflammatory responses as well as NLRP3 inflammasome-mediated IL-1β release. Consequently, these results unveil therapeutic opportunities for treatment of inflammatory diseases with Trx1 inhibitors.

## Materials and methods

**Key resources table**

| Reagent type (species) or resource | Designation | Source or reference | Identifiers | Additional information |
|---|---|---|---|---|

*Continued on next page*

*Continued*

| Reagent type (species) or resource | Designation | Source or reference | Identifiers | Additional information |
|---|---|---|---|---|
| Genetic reagent (*M. musculus*) | C57BL/6J | The Jackson Laboratory | Cat#JAX:000664; RRID: IMSR_JAX:000664 | |
| Genetic reagent (*M. musculus*) | B6 *Ptprc*[a] (CD45.1) | The Jackson Laboratory | Cat#JAX:002014, RRID:IMSR_JAX:002014 | |
| Genetic reagent (*M. musculus*) | *Txnrd1*[fl/fl] | PMID: 15713651 | | Kindly provided by Marcus Conrad |
| Genetic reagent (*M. musculus*) | Rosa26-CreERT2 | PMID: 17456738 | | |
| Genetic reagent (*M. musculus*) | *Txnip*[fl/fl] | The Jackson Laboratory | Cat# JAX:016847, RRID:IMSR_JAX:016847 | |
| Genetic reagent (*M. musculus*) | *Vav1-Cre* | PMID: 12548562 | | |
| Antibody | Anti-B220 PerCP-conjugated (rat, monoclonal) | Biolegend | Cat# 103234, RRID:AB_893353 | Flow cytometry, cell surface (1:400) |
| Antibody | Anti-CD103 FITC-conjugated (armenian hamster, monoclonal) | Biolegend | Cat# 121420, RRID:AB_10714791 | Flow cytometry, cell surface (1:200) |
| Antibody | Anti-CD11b BV605-conjugated (rat, monoclonal) | Biolegend | Cat# 101237, RRID:AB_11126744 | Flow cytometry, cell surface (1:2000) |
| Antibody | Anti-CD11b PerCP-Cy5.5-conjugated (rat, monoclonal) | Biolegend | Cat# 101228, RRID:AB_893232 | Flow cytometry, cell surface (1:1000) |
| Antibody | Anti-CD11c APC-conjugated (armenian hamster, monoclonal) | Thermo Fisher Scientific | Cat# 17-0114-82, RRID:AB_469346 | Flow cytometry, cell surface (1:300) |
| Antibody | Anti-CD11c BV605-conjugated (armenian hamster, monoclonal) | Biolegend | Cat# 117333, RRID:AB_11204262 | Flow cytometry, cell surface (1:1000) |
| Antibody | Anti-CD11c PE-Cy7-conjugated (armenian hamster, monoclonal) | Biolegend | Cat# 117318, RRID:AB_493568 | Flow cytometry, cell surface (1:500) |
| Antibody | Anti-CD19 BV650-conjugated (rat, monoclonal) | Biolegend | Cat# 115541, RRID:AB_11204087 | Flow cytometry, cell surface (1:500) |
| Antibody | Anti-CD206 PerCP-Cy5.5-conjugated (rat, monoclonal) | Biolegend | Cat# 141716, RRID:AB_2561992 | Flow cytometry, intracellular (1:400) |
| Antibody | Anti-PD-L2 PE-conjugated (rat, monoclonal) | Biolegend | Cat# 107206, RRID:AB_2162011 | Flow cytometry, cell surface (1:400) |
| Antibody | Anti-CD301b PE-Cy7-conjugated (rat, monoclonal) | Biolegend | Cat# 146808, RRID:AB_2563390 | Flow cytometry, cell surface (1:600) |
| Antibody | Anti-CD4 PerCP-Cy5.5-conjugated (rat, monoclonal) | Biolegend | Cat# 100540, RRID:AB_893326 | Flow cytometry, cell surface (1:500) |
| Antibody | Anti-CD45 BV785-conjugated (rat, monoclonal) | Biolegend | Cat# 103149, RRID:AB_2564590 | Flow cytometry, cell surface (1:1000) |
| Antibody | Anti-CD45.1 APC-conjugated (mouse, monoclonal) | Biolegend | Cat# 110714, RRID:AB_313503 | Flow cytometry, cell surface (1:100) |

*Continued on next page*

*Continued*

| Reagent type (species) or resource | Designation | Source or reference | Identifiers | Additional information |
|---|---|---|---|---|
| Antibody | Anti-CD45.1 biotin-conjugated (mouse, monoclonal) | BD Biosciences | Cat# 553774, RRID:AB_395042 | Flow cytometry, cell surface (1:200) |
| Antibody | Anti-CD45.2 APC-conjugated (mouse, monoclonal) | Thermo Fisher Scientific | Cat# 17-0454-82, RRID:AB_469400 | Flow cytometry, cell surface (1:100) |
| Antibody | Anti-CD45.2 FITC-conjugated (mouse, monoclonal) | Thermo Fisher Scientific | Cat# 11-0454-82, RRID:AB_465061 | Flow cytometry, cell surface (1:300) |
| Antibody | Anti-CD8 PE-Cy7-conjugated (rat, monoclonal) | Biolegend | Cat# 100722, RRID:AB_312761 | Flow cytometry, cell surface (1:1000) |
| Antibody | Anti-F4/80 AF488-conjugated (rat, monoclonal) | Biolegend | Cat# 123120, RRID:AB_893479 | Flow cytometry, cell surface (1:400) |
| Antibody | Anti-F4/80 BV421-conjugated (rat, monoclonal) | Biolegend | Cat# 123131, RRID:AB_10901171 | Flow cytometry, cell surface (1:300) |
| Antibody | Anti-IFN-$\gamma$ APC-conjugated (rat, monoclonal) | Biolegend | Cat# 505810, RRID:AB_315404 | Flow cytometry, intracellular (1:4000) |
| Antibody | Anti-Ly-6G BV421-conjugated (rat, monoclonal) | Biolegend | Cat# 127627, RRID:AB_10897944 | Flow cytometry, cell surface (1:800) |
| Antibody | Anti-Ly-6G PerCP-Cy5.5-conjugated (rat, monoclonal) | Biolegend | Cat# 127616, RRID:AB_1877271 | Flow cytometry, cell surface (1:600) |
| Antibody | Anti-Ly-6C APC-Cy7-conjugated (rat, monoclonal) | Biolegend | Cat# 128026, RRID:AB_10640120 | Flow cytometry, cell surface (1:2000) |
| Antibody | Anti-Ly-6C PE-Cy7-conjugated (rat, monoclonal) | Biolegend | Cat# 128018, RRID:AB_1732082 | Flow cytometry, cell surface (1:2000) |
| Antibody | Anti-MHCII PE-Cy7-conjugated (rat, monoclonal) | Biolegend | Cat# 107635, RRID:AB_2561397 | Flow cytometry, cell surface (1:400) |
| Antibody | Anti-Nos2 AF647-conjugated (mouse, monoclonal) | Santa Cruz | Cat# sc-7271, RRID:AB_627810 | Flow cytometry, intracellular (1:100) |
| Antibody | Anti-Rabbit IgG(H+L) FITC-conjugated (goat, polyclonal) | SouthernBiotech | Cat# 4055–02, RRID:AB_2795979 | Flow cytometry, Microscopy (1:500) |
| Antibody | Anti-Relm$\alpha$ (rabbit, polyclonal) | PeproTech | Cat# 500-P214bt-50ug, RRID: AB_1268707 | Flow cytometry, intracellular (1:100) |
| Antibody | Anti-Siglec-F PE-conjugated (rat, monoclonal) | BD Biosciences | Cat# 552126, RRID:AB_394341 | Flow cytometry, cell surface (1:300) |
| Antibody | Anti-TCR$\beta$ APC-conjugated (armenian hamster, monoclonal) | Biolegend | Cat# 109212, RRID:AB_313435 | Flow cytometry, cell surface (1:500) |
| Antibody | Anti-NF-$\kappa$B p65 (rabbit, polyclonal) | Biolegend | Cat# 622604, RRID:AB_2728469 | ChIP, 3 $\mu$g antibody for 40 $\mu$g chromatin |
| Antibody | CD4 MicroBeads | Miltenyi Biotec | Cat# 130-049-201, RRID:AB_2722753 | MACS enrichment |
| Antibody | CD11b MicroBeads | Miltenyi Biotec | Cat# 130-049-601, | MACS enrichment |

*Continued on next page*

*Continued*

| Reagent type (species) or resource | Designation | Source or reference | Identifiers | Additional information |
|---|---|---|---|---|
| Antibody | Anti-IL-1β (armenian hamster, monoclonal) | Thermo Fisher Scientific | Cat# 14-7012-81, RRID:AB_468396 | ELISA, coating antibody (1:150) |
| Antibody | Anti-IL-1β biotin-conjugated (rabbit, polyclonal) | Thermo Fisher Scientific | Cat# 13-7112-85, RRID:AB_466925 | ELISA, detection antibody (1:150) |
| Antibody | Anti-IL-12/IL-23 p40 (rat, monoclonal) | Thermo Fisher Scientific | Cat# 14-7125-81, RRID:AB_468444 | ELISA, coating antibody (1:250) |
| Antibody | Anti-IL-12/IL-23 p40 biotin-conjugated (rat, monoclonal) | Thermo Fisher Scientific | Cat# 13-7123-81, RRID:AB_466928 | ELISA, detection antibody (1:1000) |
| Antibody | Anti-IL-6 (rat, monoclonal) | Thermo Fisher Scientific | Cat# 16-7061-81, RRID:AB_469216 | ELISA, coating antibody (1:250) |
| Antibody | Anti-IL-6 biotin-conjugated (rat, monoclonal) | Thermo Fisher Scientific | Cat# 36-7062-85, RRID:AB_469761 | ELISA, detection antibody (1:500) |
| Antibody | Anti-TNF-α (rat, monoclonal) | BD Biosciences | Cat# 551225, RRID:AB_394102 | ELISA, coating antibody (1:100) |
| Antibody | Anti-TNF-α biotin-conjugated (rabbit, polyclonal) | Thermo Fisher Scientific | Cat# 13-7341-81, RRID:AB_466950 | ELISA, detection antibody (1:150) |
| Antibody | Anti-phospho-Erk1/2 (rabbit, polyclonal) | Cell Signaling Technology | Cat# 9101, RRID:AB_331646 | WB (1:1000) |
| Antibody | Anti-Erk1/2 (rabbit, polyclonal) | Cell Signaling Technology | Cat# 9102, RRID:AB_330744 | WB (1:1000) |
| Antibody | Anti-phospho-IκB-α, (mouse, monoclonal) | Cell Signaling Technology | Cat# 9246, RRID:AB_2267145 | WB (1:1000) |
| Antibody | Anti-IκB-α, (rabbit, polyclonal) | Cell Signaling Technology | Cat# 9242, RRID:AB_331623 | WB (1:1000) |
| Antibody | Anti-NF-κB p65 (rabbit, polyclonal) | Biolegend | Cat# 622602, RRID:AB_315956 | WB (1:2000) |
| Antibody | Anti-IL-1β (mouse, polyclonal) | R and D Systems | Cat# AF-401-NA, RRID:AB_416684 | WB (1:800), 'Mouse IL-1 beta/IL-1F2 antibody' |
| Antibody | Anti-caspase-1 p10 (rabbit, polyclonal) | Santa Cruz | Cat# sc-514, RRID:AB_2068895 | WB (1:200) |
| Antibody | Anti-β-actin peroxidase-conjugated (mouse, monoclonal) | Sigma-Aldrich | Cat# A3854, RRID:AB_262011 | WB (1:50000) |
| Antibody | Anti-NF-κB p65 (rabbit, polyclonal) | Santa Cruz | Cat# sc-7151, RRID:AB_650213 | Microscopy (1:250) |
| Peptide, recombinant protein | IFN-γ | PeproTech | Cat# 315–05 | M1 polarization (50 ng/ml) |
| Peptide, recombinant protein | IL-4 | PeproTech | Cat# 214–14 | M2 polarization (20 ng/ml) |
| Peptide, recombinant protein | M-CSF | PeproTech | Cat# 315–02 | BMDM differentiation (20 ng/ml) |
| Peptide, recombinant protein | gp61-80 peptide (LCMV) | NeoMPS | Cat# SP990990 | Sequence: GLNGPDIYKG VYQFKSVEFD |

*Continued on next page*

Continued

| Reagent type (species) or resource | Designation | Source or reference | Identifiers | Additional information |
|---|---|---|---|---|
| Chemical compound, drug | Tamoxifen (TAM) | Sigma-Aldrich | Cat# T5648-1G | To delete *Txnrd1* in *Txnrd1*$^{fl/fl}$;Rosa26-CreERT2 mice |
| Chemical compound, drug | Lipopolysaccharides (LPS) | InvivoGen | Cat# tlrl-3pelps | TLR4 agonist |
| Chemical compound, drug | Imiquimod (R837) | Tocris Bioscience | Cat# 3700 | TLR7 agonist |
| Chemical compound, drug | CpG oligodeoxynucleotides | InvivoGen | Cat# tlrl-1826–1 | TLR9 agonist |
| Chemical compound, drug | Lipoteichoic (LTA) | InvivoGen | Cat# tlrl-slta | TLR2 agonist |
| Chemical compound, drug | Zymosan A | Sigma-Aldrich | Cat# Z4250-250MG | TLR2/Dectin-1 agonist |
| Chemical compound, drug | Nigericin (Nig) | Sigma-Aldrich | Cat# N7143 | NLRP3 inflammasome inducer |
| Chemical compound, drug | Adenosine triphosphate (ATP) | Sigma-Aldrich | Cat# A7699 | NLRP3 inflammasome inducer |
| Chemical compound, drug | Alu-Gel-S (Alum) | SERVA Electrophoresis | Cat# 12261.01 | NLRP3 inflammasome inducer |
| Chemical compound, drug | Monosodium urate (MSU) crystals | InvivoGen | Cat# tlrl-msu-25 | NLRP3 inflammasome inducer |
| Chemical compound, drug | DL-Buthionine-(*S*,*R*)-sulfoximine (BSO) | Sigma-Aldrich | Cat# B2640-500MG | Glutathione synthesis inhibitor |
| Chemical compound, drug | Catalase-polyethylene glycol (Cat) | Sigma-Aldrich | Cat# C4963 | Antioxidant |
| Chemical compound, drug | L-Ascorbic acid | Sigma-Aldrich | Cat# A92902-25G | Antioxidant |
| Chemical compound, drug | Diphenyleneiodonium chloride (DPI) | Sigma-Aldrich | Cat# D2926-10MG | Antioxidant |
| Chemical compound, drug | DL-Dithiothreitol (DTT) | Sigma-Aldrich | Cat# 43815–5G | Antioxidant |
| Chemical compound, drug | N-Acetyl-L-Cysteine (NAC) | Sigma-Aldrich | Cat# A7250-10G | Antioxidant |
| Chemical compound, drug | DNase I | Sigma-Aldrich | Cat# 4716728001 | Tissue digestion (0.02 mg/ml) |
| Chemical compound, drug | Collagenase IV | Worthington | Cat# LS004189 | Tissue digestion (2 mg/ml) |

*Continued on next page*

*Continued*

| Reagent type (species) or resource | Designation | Source or reference | Identifiers | Additional information |
|---|---|---|---|---|
| Chemical compound, drug | Phorbol 12-myristate 13-acetate (PMA) | Sigma-Aldrich | Cat# P-8139 | T-cell restimulation ($10^{-7}$ mol/L) |
| Chemical compound, drug | Ionomycin | Sigma-Aldrich | Cat# I-0634 | T-cell restimulation (1 µg/ml) |
| Chemical compound, drug | Monensin | Sigma-Aldrich | Cat# M5273 | Used at 2 µg/ml during T-cell restimulation |
| Commercial assay or kit | NF-κB p65 Transcription Factor Assay Kit | Abcam | Cat# ab133112 | |
| Commercial assay or kit | Nuclear Extraction Kit | Abcam | Cat# ab113474 | |
| Commercial assay or kit | Glutathione Assay Kit | Sigma-Aldrich | Cat# CS0260 | |
| Commercial assay or kit | Pierce BCA Protein Assay Kit | Thermo Fisher Scientific | Cat# 23225 | |
| Commercial assay or kit | Pierce LDH Cytotoxicity Assay Kit | Thermo Fisher Scientific | Cat# 88954 | |
| Software, algorithm | FlowJo Software (version 10.4.2) | Three Star | https://www.flowjo.com/ | |
| Software, algorithm | ImageJ (for image analysis) | NIH | https://imagej.nih.gov/ij/ | |
| Software, algorithm | Prism 8 (version 8.0.0) | GraphPad Software | https://www.graphpad.com/scientific-software/prism/ | |
| Other | DAPI | Sigma-Aldrich | Cat# D9542 | Microscopy, nuclei visualization (1:5000) |
| Other | Phalloidin, AF-647-conjugated | Thermo Fisher Scientific | Cat# A22287 | Microscopy, actin staining (1:1000) |
| Other | eFluor 780 | Thermo Fisher Scientific | Cat# 65-0865-14 | Live/Dead staining (1:2000) |
| Other | Zombie Aqua | Biolegend | Cat# 423101 | Live/Dead staining (1:400) |
| Other | Zombie Red | Biolegend | Cat# 423109 | Live/Dead staining (1:800) |
| Other | CM-H$_2$DCFDA | Thermo Fisher Scientific | Cat# C6827 | General oxidative stress indicator (1:1000) |
| Other | Annexin-V APC-conjugated | Thermo Fisher Scientific | Cat# 88-8007-74 | Apoptosis detection (1:50) |
| Other | Streptavidin BV711-conjugated | BD Biosciences | Cat# 563262 | (1:1000) |
| Other | Dynabeads Protein G | Thermo Fisher Scientific | Cat# 10004D | For the ChIP experiment |

## Mice

*Txnrd1*$^{fl/fl}$ mice (*Jakupoglu et al., 2005*) were provided by Marcus Conrad (Helmholtz Zentrum, Munich, Germany) and were backcrossed for more than eight generations to C57BL/6. To generate *Txnrd1*$^{fl/fl}$;Rosa26-CreERT2 mice, *Txnrd1*$^{fl/fl}$ mice were crossed with Rosa26-CreERT2 (*Hameyer et al., 2007*) mice. *Txnip*$^{fl/fl}$ mice (*Yoshioka et al., 2007*) were purchased from The

Jackson Laboratory (Bar Harbor, Maine, USA) and crossed with *B6.Cg-Tg(Vav1-Cre)A2Kio/J (Vav1-Cre)* mice (*de Boer et al., 2003*). B6 *Ptprca* (CD45.1) animals were also purchased from The Jackson Laboratory (Bar Harbor, Maine, USA). About 6–12 week-old age- and sex-matched mice (either female or male) were used for the experiments. Mice were kept in individually ventilated cages under specific pathogen free conditions at the ETH Phenomics Center (EPIC; Zurich, Switzerland). All animal experiments were approved by the local animal ethics committee (Kantonales Veterinärsamt Zürich) and were performed according to local guidelines (TschV, Zurich) and the Swiss animal protection law (TschG). For deletion of the *Txnrd1* gene in *Txnrd1*$^{fl/fl}$;Rosa26-CreERT2 mice, animals were intraperitoneally injected with 2 mg TAM (Sigma-Aldrich) on two consecutive days and used for experiments at least 10 days later. For in vivo GSH depletion, L-buthionine-(S,R)-sulfoximine (BSO; Sigma-Aldrich) was supplemented in the drinking water at a concentration of 20 mM for the indicated time.

## Bone marrow chimeras

For bone marrow chimeras, C57BL/6 (CD45.1$^+$CD45.2$^+$) recipients were irradiated twice with 4.75 Gy with a 4 hr break in a RS 2000 (Rad Source Technologies Inc, Alpharetta, USA). The following day, mice were reconstituted by intravenous injection of $1-3 \times 10^6$ bone marrow cells of the donor mice (1 to 1 mixture of bone marrow from CD45.1$^+$ and CD45.2$^+$ donors). Animals were analyzed 8–10 weeks after reconstitution. Either *Txnrd1*$^{fl/fl}$;Rosa26-CreERT2 (and *Txnrd1*$^{fl/fl}$ as control) donor mice or CD45.1$^+$CD45.2$^+$ recipient mice after reconstitution were intraperitoneally treated twice with 2 mg TAM to delete *Txnrd1* gene.

## Emergency hematopoiesis

Mice were intraperitoneally injected twice with 35 µg LPS (Ultrapure *E. coli* 0111:B4, InvivoGen) in a 48 hr interval and were sacrificed for analysis 24 hr later. Bone marrow and blood were then taken and processed for analysis.

## MSU-induced peritonitis

Mice were intraperitoneally injected with 1.8 mg MSU crystals (InvivoGen) dissolved in 0.2 ml sterile PBS. After 8 hr, mice were euthanized by $CO_2$ exposure, and peritoneal cavities were washed with 10 ml PBS + 2 mM EDTA (Sigma-Aldrich). Neutrophil infiltration in peritoneal lavage fluids was subsequently assessed by CD11b and Ly-6G flow cytometry staining, whereas concentration of IL-1β was determined by ELISA.

## Cell-suspension preparations

Mice were sacrificed by an intraperitoneal overdose of sodium pentobarbital. Organs were removed and then processed according to the following procedure. Lungs and spleens were minced and then digested for 45 min at 37°C in IMDM medium (Life Technologies) containing 2 mg/ml of type IV collagenase (Worthington) and 0.02 mg/ml DNaseI (Sigma). All other organs were directly disrupted and passed through a 70 µm cell strainer (Corning). Bone marrow cells were flushed from femurs and tibia, and then directly passed through the 70 µm cell strainer. Leukocytes from the liver were isolated by using Percoll gradient centrifugation (GE Healthcare). ACK buffer was used for erythrocyte lysis for all organs.

## Cell culture

Bone-marrow cells from femur and tibia of sex-matched 6–12 week-old mice were differentiated into BMDCs in RPMI-1640 medium (Gibco) supplemented with GM-CSF (supernatant from X63-GMCSF cell line), 2 mM L-glutamine (GE Healthcare), 10 mM HEPES (Lonza), 100 U/mL penicillin, 100 µg/mL streptomycin (Gibco), 10% FCS (Gibco). Fresh medium was added on day 3 and day 6 of culture, and non-adherent cells were harvested and used in experiments on day 7 of culture. BMDMs were cultured analogously, but with medium supplemented with 20 ng/ml recombinant M-CSF (PeproTech). Adherent BMDMs detached by washing plates with cold PBS + 2 mM EDTA (Sigma-Aldrich) were used in experiments. For differentiation of BMDMs toward M1 and M2, cells were polarized in the presence of ultra-pure LPS from *Escherichia coli O111:B4* (20 ng/ml; InvivoGen) + IFN-γ (50 ng/ml; PeproTech) and IL-4 (20 ng/ml; PeproTech), respectively. For analysis of M1/M2 markers at the

RNA and protein levels, BMDMs were polarized for 24 and 48 hr, respectively. For BMDC/BMDM stimulation, cells were generally primed with R837 (Tocris Bioscience), CpG (InvivoGen), Ultra-pure LPS from *Escherichia coli O111:B4* (InvivoGen), lipoteichoic acid from *Staphylococcus aureus* (InvivoGen), and zymosan A (Sigma-Aldrich). Cells were generally primed with the indicated TLR ligands for 3 hr before stimulation for 4 hr with 200 µg/ml alum ($Al(OH)_3$; SERVA Electrophoresis), or for 6 hr before stimulation for 1 hr with 2 mM adenosine 5'-triphosphate disodium salt (ATP; Sigma-Aldrich) or 5 µM nigericin sodium salt (Sigma-Aldrich) or as indicated in each figure legend. Other compounds used were applied overnight and consisted of the following: L-ascorbic acid (Sigma-Aldrich), L-buthionine-sulfoximine (BSO; Sigma-Aldrich), catalase-polyethylene glycol (catalase-PEG; Sigma-Aldrich), diphenyleneiodonium chloride (DPI; Sigma-Aldrich), DL-dithiothreitol (DTT; Sigma-Aldrich), and N-acetyl-L-Cysteine (NAC; Sigma-Aldrich). The concentrations at which all the compounds were used are indicated in each figure legend.

## In vitro CD4+ T cell-DC co-culture

$5 \times 10^3$ *Txnrd1*-deficient and sufficient BMDCs were co-cultured with GP61-80-specific, MACS-enriched naïve CD4+ T cells ($25 \times 10^3$/well) in the presence of the indicated doses of GP61-80 peptide (GLNGPDIYKGVYQFKSVEFD, I-A^b-restricted). Co-cultures were performed in IMDM + Gluta-MAX, 10% FCS, 100 U/ml penicillin, 100 µg/ml streptomycin, 50 µM β-mercaptoethanol (all Gibco). After 4 days, CD4+ T cells were analyzed for the intracellular expression of IFN-γ upon restimulation with phorbol 12-myristate 13-acetate (PMA; $10^{-7}$ mol/L; Sigma) + ionomycin (1 µg/ml; Sigma) in the presence of monensin (2 µg/ml; Sigma) for 4 hr at 37°C.

## Flow cytometry

For dead cell exclusion, cells were stained with either eFluor 780 (eBioscience) or with the Zombie Aqua or Red Fixable Viability kits (Biolegend). Prior to surface staining with antibodies, Fc gamma receptors were blocked by incubating cells with anti-CD16/CD32 antibody (2.4G2, homemade). Antibodies for extracellular stains were then incubated with cells for 15 min in FACS buffer (PBS + 2% FCS). When intracellular staining was required (for IFN-γ, Relmα, CD206 and Nos2 stainings), cells were additionally fixed with 4% formalin, permeabilized with PBS supplemented with 2% FCS and 0.5% saponin (permeabilization buffer), and intracellularly stained for 30 min in permeabilization buffer. The anti-Relmα antibody was detected with an anti-rabbit secondary antibody labeled with the FITC fluorophore. To analyze cell death, cells were stained with Annexin-V-APC (BD Bioscience) and eFluor 780 (eBioscience) in Annexin-V binding buffer. Total cellular ROS were quantified by CM-H₂DCFDA staining (Life Technologies) and measuring fluorescein signal after 30 min recovery in supplemented medium. Cells were acquired on LSRFortessa (BD Bioscience), or sorted on FACSAria III (BD Bioscience). Data were analyzed in FlowJo software (Tree Star). All antibodies and staining reagents with their respective dilutions used in this study can be found in the *Key resources table*.

## DNA binding activity of NF-κB p65

To measure the NF-κB p65 transcription factor DNA binding activity, nuclear extracts from LPS-primed BMDCs or BMDMs were first isolated using the Nuclear Extraction Kit (Abcam), and the NF-κB binding activity was subsequently measured using the NF-κB p65 Transcription Factor Assay Kit (Abcam). Both kits were used following the manufacturer's instructions. Values obtained from the Transcription Factor Assay Kit were normalized for the total protein concentrations in the nuclear extracts, which were determined using the Pierce BCA Protein Assay Kit (Thermo Scientific).

## Magnetic cell sorting

CD11b+ and CD4+ enrichments were achieved by positive selection using a MACS system with microbeads conjugated to monoclonal anti-mouse CD11b and CD4 antibodies (MACS, Miltenyi Biotec), respectively, following the manufacturer's instructions.

## Glutathione measurement

To measure total GSH levels in cell lysates, the Glutathione Assay Kit (Sigma-Aldrich) was used following the manufacturer's instructions.

## ELISA

Cytokines in supernatants were quantified by sandwich ELISA using the following pairs of capture and detection antibodies: B122 and 13-7112-85 (eBioscience) for IL-1β, C15.6 and C17.8 (eBioscience) for IL-12p40, MP5-20F3 and MP5-32C11 (Biolegend) for IL-6, and G281-2626 (BD Pharmingen) and biotinylated anti-TNF-α polyclonal antibody (13-7341-85; eBioscience) for TNF-α. After the coating and detection steps, alkaline phosphatase (AP)-conjugated streptavidin (SouthernBiotech) was provided, and the alkaline phosphatase p-nitrophenyl phosphate (pNPP; Sigma-Aldrich) substrate was subsequently added to each well. The plates were finally read at 405 nm.

## RNA analysis by real-time quantitative PCR

Total RNA was extracted using TRIzol (Life Technologies), followed by reverse transcription using GoScript Reverse Transcriptase (Promega) according to the manufacturer's instructions. Real-time quantitative PCR (RT-PCR) was performed using Brilliant SYBR Green (Stratagene) on an i-Cycler (Bio-Rad Laboratories) according to manufacturer's protocol. Expression was normalized to the housekeeping gene *Tbp* for mRNA expression, or to genomic *Txnrd1* for addressing DNA recombination efficiency in cells lacking the *Txnrd1* gene. The sequences of all used primers are listed in *Supplementary file 1*.

## Chromatin immunoprecipitation (ChIP)

$5 \times 10^6$ BMDCs per genotype were fixed with 1% formaldehyde (10 min at 37°C), and cross-linking was stopped by adding glycine to a final concentration of 0.125 M. Cells were washed using ice-cold PBS, resuspended in swelling buffer (25 mM HEPES [pH 8], 1.5 mM MgCl$_2$, 10 mM KCl, 0.5% NP-40) with protease inhibitors for 15 min, spun down and finally resuspended in 400 µl lysis buffer (PBS [1x], 1% SDS, 1% NP-40, 0.5% sodium deoxycholate) with protease inhibitors for 10 min. Sonication was subsequently performed using the Bioruptor (Diagenode; 30 cycles of 30 s on, 30 s off) at 4°C to shear chromatin into ca. 200–300 base-pair DNA fragments. Sonicated samples were diluted with 1.6 ml of lysis buffer without SDS (PBS [1x], 1% NP-40, 0.5% sodium deoxycholate). 40 µg sonicated chromatin in 1 ml volume was precleared by adding 50 µl of dynabeads (Thermo Scientific) and incubation on a wheel for 2 hr at 4°C. At this stage, 10% of chromatin was taken as input DNA. Supernatant was then collected, 3 µg of p65 antibody (Biolegend) was added, and samples were incubated overnight at 4°C on a wheel. Chromatin-antibody complexes were then retrieved by adding 50 µl of dynabeads (Thermo Scientific) and incubation for 4 hr at 4°C on a wheel. Beads were washed once with ice-cold low-salt wash buffer (16.7 mM Tris-HCl [pH 8], 0.167 M NaCl, 0.1% SDS, 1% Triton-X), once with ice-cold high-salt wash buffer (16.7 mM Tris-HCl [pH 8], 0.5 M NaCl, 0.1% SDS, 1% Triton-X), twice using LiCl wash buffer (0.25 M LiCl, 10 mM Tris-HCl [pH 8], 1 mM EDTA [pH 8], 0.5% sodium deoxycholate, 0.5% NP40), and once with TE buffer (10 mM Tris-HCl [pH 8], 5 mM EDTA [pH 8]). DNA was then eluted in elution buffer (1% SDS, 100 mM NaHCO$_3$) by gentle shaking for 30 min at 37°C, and cross-links were reversed by overnight incubation at 65°C. After RNase A and proteinase K treatment, DNA was purified via phenol/chloroform extraction and ethanol precipitation.

## Immunoblotting

Cells were lysed on ice with RIPA buffer (20 mM Tris-HCl, pH 7.5, 150 mM NaCl, 5 mM EDTA, 1 mM Na$_3$VO$_4$, 1% Triton X-100, supplemented with protease inhibitor [Sigma-Aldrich] and phosphatase inhibitor [Sigma-Aldrich]). Cell debris was then removed by spinning for 10 minutes at 4°C. Protein concentrations were determined using the Pierce BCA Protein Assay Kit (Thermo Scientific). Whole cell extracts (30 µg of proteins) were fractionated by SDS-PAGE and transferred to a polyvinylidene difluoride (PVDF) membrane using a transfer apparatus according to manufacturer's instructions (Bio-Rad). Membranes were blocked with 4% nonfat milk in TBST (50 mM Tris, pH 8.0, 150 mM NaCl, 0.1% Tween20) for 45 minutes, washed and incubated with primary antibodies (1:1000 in TBST with 4% BSA) at 4°C for 12 hours. After washing, membranes were incubated with a 1:2000 dilution (in TBST with 4% nonfat milk) of horseradish peroxidase-conjugated anti-rabbit or anti-mouse antibodies for 1 hour. Blots were washed with TBST three times and developed with the ECL system (Thermo Scientific) according to manufacturer's instructions. All antibodies for immunoblotting used in this study can be found in the Key resources table.

## Immunofluorescence analysis

BMDCs were stimulated with LPS (400 ng/ml) on glass coverslips for the indicated times. After stimulation, cells were fixed in 2% paraformaldehyde and subsequently permeabilized in 0.25% Triton X-100 in PBS. For staining, cells were incubated with anti-NF-κB p65 antibody (Santa Cruz) in 0.1% Triton X-100 containing 2% goat serum in PBS. Secondary antibodies were anti-rabbit IgG, ads-FITC (Southern Biotech). During incubation with the secondary antibody, cells were additionally stained for DNA with DAPI (Sigma-Aldrich) and for actin with phalloidin-647 (Thermo Scientific). The coverslips were mounted on glass slides with Vectashield mounting medium (AdipoGen) for immunofluorescence analysis. Microscopy was performed using an oil immersion objective (UPlanSApo, Olympus; Magnification 100×, NA = 1.4) on a DeltaVision microscope system coupled to a sCMOS camera (pco.edge 5.5, PCO). For quantification of NF-κB nuclear translocation, DAPI and actin stainings were used to create masks for nuclei and whole cell. NF-κB signal intensity within these masks were separately quantified.

## Statistical analysis

Data were analyzed using either a Student's *t* test (two-tailed, unpaired), or one-way ANOVA followed by either Tukey's or Dunnett's corrections, or a two-way ANOVA adjusted by Bonferroni's multiple comparison test. The data are represented as mean + standard deviation. The method of statistical evaluation and the significance levels are also described in each figure legend.

## Acknowledgements

We are grateful for research grants from ETH Zurich (ETH-23-16-2) and SNF (310030B_182829). We thank Marcus Conrad for providing *Txnrd1*$^{fl/fl}$ mice. We thank Peter Nielsen for designing primers and Madlen Müller for help during the ChIP experiment. We further thank members of the ETH Flow Cytometry Core Facility for cell sorting. We acknowledge the use of the Immgen database as an informative tool for our study.

## Additional information

### Funding

| Funder | Grant reference number | Author |
| --- | --- | --- |
| ETH Zürich | ETH-23-16-2 | Manfred Kopf |
| Swiss National Science Foundation | 310030B_182829 | Manfred Kopf |

The funders had no role in study design, data collection and interpretation, or the decision to submit the work for publication.

### Author contributions

Jonathan Muri, Conceptualization, Data curation, Formal analysis, Investigation, Visualization, Methodology, Writing - original draft, Writing - review and editing; Helen Thut, Data curation, Formal analysis, Investigation; Qian Feng, Data curation, Software, Formal analysis, Investigation, Visualization, Methodology; Manfred Kopf, Conceptualization, Supervision, Funding acquisition, Writing - original draft, Writing - review and editing

### Author ORCIDs

Jonathan Muri ◉ https://orcid.org/0000-0002-6476-3766
Manfred Kopf ◉ https://orcid.org/0000-0002-0628-7140

### Ethics

Animal experimentation: All animal experiments were approved by the local animal ethics committee (Kantonales Veterinärsamt Zürich, licenses 25/2014, ZH054/18, ZH054/19 and ZH134/18), and performed according to local guidelines (TschV, Zurich) and the Swiss animal protection law (TschG).

Decision letter and Author response

Decision letter https://doi.org/10.7554/eLife.53627.sa1

Author response https://doi.org/10.7554/eLife.53627.sa2

## Additional files

### Supplementary files

- Supplementary file 1. Sequences of primers used for RT-PCR.

- Transparent reporting form

### Data availability

All data generated or analysed during this study are included in the manuscript and supporting files.

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
