## [Decision Letter]

**Acceptance summary:**

Antioxidant systems maintain cellular redox homeostasis, and this study showed that TrxR1, a key enzyme in the Thioredoxin system, influences emergency myelopoiesis, inflammatory gene induction in dendritic cells, and NLRP3 inflammasome activation. The findings indicated that TrxR1 activity modulates an optimal level of cellular ROS that is needed for normal myeloid cell functionality.

**Decision letter after peer review:**

Thank you for submitting your article "Thioredoxin-1 differentially promotes NF-κB target DNA binding and NLRP3 inflammasome activation independently of Txnip" for consideration by *eLife*. Your article has been reviewed by three peer reviewers, one of whom is a member of our Board of Reviewing Editors, and the evaluation has been overseen by Satyajit Rath as the Senior Editor. The reviewers have opted to remain anonymous.

The reviewers have discussed the reviews with one another and the Reviewing Editor has drafted this decision to help you prepare a revised submission.

Summary:

Antioxidant systems maintain cellular redox homeostasis, yet how this affects myeloid cell biology is not well understood. Here Kopek et al. examine the effects of genetic deletion of *Txnrd1*, a key enzyme in the Thioredoxin system, on myeloid cell development/homeostasis, polarization, and effector functions.

The authors found that *Txnrd1* deletion had little effect on steady state myeloid cell numbers, but led to significant reduction in a competitive setting with WT counterparts and during emergency myelopoiesis.

Upon exposure to Toll-like receptor agonists, *Txnrd1*-deficient bone marrow derived dendritic cells (BMDCs) displayed reduced induction of Il12p40, Il1b, and Il6, which was attributed to reduced p65 DNA binding. However, *Txnrd1*-deficient bone marrow derived macrophages (BMDMs) compensated by upregulating enzymes of the GSH/Grx system leading to no reduction of inflammatory gene induction.

Finally, the authors examined NLRP3 inflammasome activation in BMDMs. The authors found that *Txnrd1* deficiency led to decreased NLRP3 inflammasome activation, which they attributed to increased ROS production, since treatment with cell permeable catalase rescued the defect.

Essential revisions:

1) The authors should obtain more support for their conclusion that excessive levels of ROS impair NLRP3 inflammasome activation given that many previous studies have reached opposite conclusions.

• The authors are advised to more definitely examine NLRP3 inflammasome activation by interrogating IL-1β and caspase-1 processing by western blot.

• The authors need to offer some explanation, ideally supported by experimental evidence, for the discrepant role of ROS in inflammasome activation in their data versus the literature. For example, is it possible to gradually titrate the amount of catalase added to either WT or *Txnrd1* deficient BMDMs and to see dose dependent effects on DCFDA staining concomitant with different effects on inflammasome activation? In Figure 7H-K, can the authors confirm that addition of catalase does not affect Il1b gene induction? At the very least, the authors should propose experiments that could be done in future studies to confirm the differing effects of excessive versus moderate ROS levels on inflammasome activation.

2) Clarification is needed on a few points in the second part of the story addressing the role of the Trx system in NF-κB activation.

• The authors should confirm the defect in p65 DNA binding using p65 ChIP at the promoters of the inflammatory genes.

• What is the proposed mechanism for why *Txnrd1* deficiency is more specific for NF-κB regulation than general oxidation? A control that demonstrates the general efficacy of the reducing agents seems warranted.

• It is not clear why nuclear NF-κB is unchanged (Figure 5B) but DNA binding is affected (Figure 5E) in the KO.

3) In the first part of the story, the authors conclude that reduced myeloid cell numbers in the absence of the antioxidant system(s) is due to defects in proliferation, but data to support this is lacking. The authors are advised to describe the data as defects in "maintenance" and "homeostasis".

---

## [Author Response]

Essential revisions:1) The authors should obtain more support for their conclusion that excessive levels of ROS impair NLRP3 inflammasome activation given that many previous studies have reached opposite conclusions.• The authors are advised to more definitely examine NLRP3 inflammasome activation by interrogating IL-1β and caspase-1 processing by western blot.

We thank the reviewers for this important comment. As suggested, we have now performed western blotting for IL-1β and caspase-1 in both the cell lysate and SN upon stimulation with LPS/R837 and ATP/Alum to interrogate NLRP3 inflammasome activation. Consistent with the impaired secretion of IL-1β in the absence of the Trx1 system measured by ELISA, we found reduced levels of both processed IL-1β and caspase-1 by western blot (see new Figure 7C and Figure 7—figure supplement 1B). Thus, these data are in line with a defect in the activation of the NLRP3 inflammasome when the *Txnrd1* gene is deleted.

• The authors need to offer some explanation, ideally supported by experimental evidence, for the discrepant role of ROS in inflammasome activation in their data versus the literature. For example, is it possible to gradually titrate the amount of catalase added to either WT or Txnrd1 deficient BMDMs and to see dose dependent effects on DCFDA staining concomitant with different effects on inflammasome activation?

We agree with the reviewers that this is a critical point. First of all, it is important to mention that, as already described in the Discussion section, nitrogen oxide (a key nitrogen reactive species [RNS]) can nitrosylate NLRP3 and caspase-1, thus leading to NLRP3 inflammasome inhibition (see for example PMID: 9780184 and PMID: 23160153). One could imagine a similar mechanism with ROS and in particular with H_2_O_2_, where excessive level of this particular ROS may result in the oxidation/peroxidation of inflammasome components (such as ASC, NLRP3, and caspase-1 among others) or other inflammasome regulators, thus compromising their proper functionality. This would be consistent with the defect that we have observed in IL-1β and caspase-1 processing. We believe that our data/conclusions are not opposite to other studies in the field as we report that only excessive levels of H_2_O_2_ due to *Txnrd1* deletion impair inflammasome function. To further strengthen this model, we followed the suggestions of the reviewers and titrated the amount of catalase to see dose dependent effects on DCFDA staining and IL-1β secretion (see new Figures 7I and 7J). Interestingly, we now show that only low levels of catalase could rescue IL-1β production in the absence on *Txnrd1* by bringing ROS to the levels observed in WT cells. By contrast, increased catalase concentrations, which led to a more drastic inhibition of ROS, turned out to display the opposite effects and thus inhibit inflammasome formation in both WT and KO BMDMs. This observation is in line with the many reports in the literature that describe ROS as inducers of NLRP3 inflammasome. In conclusion, our data confirms that ROS are involved in NLRP3 inflammasome activation; however, they also highlight that an excess of ROS (for example due to the genetic ablation of the key antioxidant pathway Trx1) impairs NLRP3 activation. As mentioned, the underlying molecular mechanism is unknown and currently under investigation, but it could involve a general oxidation of inflammasome components which compromises their proper functionality.

In Figure 7H-K, can the authors confirm that addition of catalase does not affect Il1b gene induction?

We are thankful for the reviewers’ thoughtful suggestion. First of all, it is important to mention here that the levels of IL-1β mRNA in KO BMDMs, as we have already shown in our initial submission (see new Figure 7B), are higher compared to WT cells, despite a large impairment in IL-1β secretion (probably due to compensation mechanisms). Importantly, however, by performing the experiment suggested by the reviewers, we have observed that catalase treatment does not affect the transcriptional levels of IL-1β (after either LPS or CpG priming) in both WT and KO cells (see new Figure 7—figure supplement 3). Thus, these data suggest that catalase treatment promotes inflammasome activation rather than affecting priming in KO BMDMs.

At the very least, the authors should propose experiments that could be done in future studies to confirm the differing effects of excessive versus moderate ROS levels on inflammasome activation.

To date, various cellular and molecular events have been proposed as trigger for the NLRP3 inflammasome, such as potassium efflux, lysosomal destabilization and ROS among others. Although ROS are one of the first NLRP3 activators that have been identified, the exact mechanism by which the NLRP3 inflammasome senses ROS remains still unknown and is beyond the aim of this study. A clear example for the huge controversy of ROS in the inflammasome field is given by the role of the key antioxidant pathway Nrf2. In particular, Nrf2 has been associated with both a negative (e.g. PMID 27211851; PMID 29590092) and a positive (e.g. PMID: 24798340; PMID: 21484785) regulation of IL-1β responses. Thus, we strongly believe that understanding how ROS affect NLRP3 inflammasome activation and especially how various concentrations of ROS have different effects might be not so simple. As we have already mentioned before, we believe and our data suggest that moderate/medium ROS levels trigger or participate in the activation of the NLRP3 inflammasome, while excessive/damaging ROS concentrations are inhibitory because they may oxidize and compromise the functionality of factors that are needed for inflammasome formation. To test this in future studies, we are currently planning to determine protein oxidations caused by *Txnrd1* deficiency using unbiased mass spectrometry. Likely candidates might be oxidations on cysteine (Cys-disulfide; Cys-sulfenic acid; Cys-sulfinic acid; Cys-sulfonic acid) and on methionine (Met-sulfoxide), which are generally caused by oxidizing agents such as ROS (this has now been added to the Discussion section). One should for example analyze the most significant differences in oxidation patterns between WT and KO cells, focusing on known inflammasome components and other regulators. Using this strategy, we actually aim to understand whether genetic ablation of a key antioxidant pathway leading to high and detrimental ROS levels shifts the redox balance and leads to oxidation of proteins involved in NLRP3 inflammasome function.

2) Clarification is needed on a few points in the second part of the story addressing the role of the Trx system in NF-κB activation.• The authors should confirm the defect in p65 DNA binding using p65 ChIP at the promoters of the inflammatory genes.

We agree with the reviewers that validating the results obtained with the ELISA-based method will strengthen the conclusion. Indeed, as suggested, we have now performed p65 ChIP-qPCR at the promoters of the inflammatory genes (the details about the utilized procedure can be found in the new method section). We tested four different genes (*IL-12p40*, *IL-6*, *IL-1β* and *Nfkbia*), which are all well-known NF-κB target genes, and we performed qPCR utilizing two different primer sets for each gene. The first primer pair (named as “promoter”) allowed us to amplify a fragment close to the NF-κB binding sites at the promoter region, whereas the second primer pair (named as “Ctr primers”) was used as a control to amplify a region several kilobases away from the NF-κB binding sites. As depicted in the new Figure 5D, we were able to confirm that *Txnrd1*-deficient BMDCs display a reduced NF-κB binding to target DNA.

• What is the proposed mechanism for why Txnrd1 deficiency is more specific for NF-κB regulation than general oxidation? A control that demonstrates the general efficacy of the reducing agents seems warranted.

We thank the reviewers for mentioning this important point. Unfortunately, it seems like we were not clear enough in our initial description of the data, which must have resulted in a misunderstanding of the reviewer(s). The reason why we studied the NLRP3 inflammasome phenotype exclusively in BMDMs is that they do not display any defects in IL-1β transcription. In contrast, BMDCs have a drastic impairment in TLR priming, thus rendering the investigation of NLRP3-inflammasome function more complicated. This said, our data do not indicate that “Txnrd1 deficiency is more specific for NF-κB regulation than general oxidation” in BMDCs; indeed, in BMDCs, we observed both a defect in NF-κB binding to target DNA and a general increase of ROS, which is likely leading to a defective NLRP3 activation as we have shown for BMDMs. Originally, we omitted these data from our manuscript to reduce complexity and facilitate readability and understanding of the story. Unfortunately, it seems like that we reached the opposite and created confusion of the reviewer(s). Therefore, we have now included these data in the new version of the manuscript (see new Figure 5—figure supplement 3). Here, we show that BMDCs also display higher ROS levels in the absence of *Txnrd1*, and that catalase-treatment, although it inhibits ROS formation, cannot restore cytokine production. Therefore, it looks like both BMDCs and BMDMs require Txnrd1 to inhibit ROS and allow inflammasome formation; by contrast, the NF-κB phenotype is exclusively observed in BMDCs, as BMDMs express high levels of glutaredoxins which can compensate for the absence of the Trx1 system. It is additionally important to mention that the binding of NF-κB to the target DNA is not ROS dependent (as catalase, which blocks ROS production, does not restore cytokine production) but specifically requires the Trx1 protein (or Grx as a backup), for example by providing reducing power for the regulation of NF-κB redox state (e.g. disulfide bond formation). This is actually reminiscent of what we have previously observed in T cells, where the enzymatic activity of Trx1, but not its general antioxidant function, was needed to deliver electrons to a specific protein, namely ribonucleotide reductase for the synthesis of dNTPs.

• It is not clear why nuclear NF-κB is unchanged (Figure 5B) but DNA binding is affected (Figure 5E) in the KO.

We thank the reviewers for raising this point and we are sorry for the lack of clarity. NF-κB induced inflammatory responses can be separated into at least three steps including (i) cytoplasmic NF-κB activation, (ii) nuclear translocation, and (iii) DNA binding followed by transcription of target genes, which are controlled by different biochemical mechanisms. We found reduced production of pro-inflammatory cytokines known to be controlled NF-κB in *Txnrd1*-deficient BMDCs. We showed that cytoplasmic NF-κB activation and nuclear translocation were unaffected indicating that DNA binding, which is differentially regulated, is impaired in the absence of *Txnrd1*. Indeed, in the first submitted version of the manuscript, we showed that DNA binding is impaired in *Txnrd1*-deficient BMDCs using a binding assay. In the revised version, as requested by the reviewers, we supported this finding by performing p65-CHIP analysis of NF-κB target genes.

What it actually remains to be answered in future studies would be why and how the p65 DNA binding is affected. In this regard, we believe that Trx1 controls and regulates the redox status of p65 (and possibly of other factors that participate in DNA binding). For instance, Trx1 may provide electrons to reduce disulfide bonds of p65, and this may promote the DNA binding activity of the transcription factor. By using mass spectrometry, one could better analyze and understand how exactly the Trx1 system regulates the redox state of p65.

3) In the first part of the story, the authors conclude that reduced myeloid cell numbers in the absence of the antioxidant system(s) is due to defects in proliferation, but data to support this is lacking. The authors are advised to describe the data as defects in "maintenance" and "homeostasis".

We thank the reviewers for pointing out this aspect that was not fully supported by the data in our manuscript. We have previously described the defect as a “proliferation defect” since this is what we have originally reported in the T cells (where the Trx1 system donates electrons for the synthesis of DNA building blocks during rapid proliferation; PMID: 29749372). Although we believe that a similar mechanism very likely also takes place in myeloid cells, we agree with the reviewers that the data in the current manuscript do not unambiguously prove it. Thus, we have now followed the reviewers’ suggestion and described the data in this manuscript as defects in “maintenance” and “homeostasis”.